



# Compositional data analysis (CoDA) as a tool to evaluate a new low-cost settling-based PM₁₀ sampling head in a desert dust source region

Yangjunjie Xu-Yang[1], Rémi Losno[1], Fabrice Monna[2], Jean-Louis Rajot[3,4], Mohamed Labiadh[5], Gilles Bergametti[3], and Béatrice Marticorena[3]

[1]Université de Paris, Institut de Physique du Globe de Paris, UMR CNRS 7154, Paris, France
[2]ARTEHIS, UMR CNRS 6298, Université Bourgogne-Franche Comté, Dijon, France
[3]LISA, Université Paris Est Créteil, Université de Paris, UMR CNRS 7583, France
[4]IRD, Institut pour la Recherche et le développement
[5]Institut des Régions Arides, Medenine, Tunisia

**Correspondence:** Rémi Losno (losno@ipgp.fr)

**Abstract.** This paper presents a new sampling head design and the method used to evaluate it. The elemental composition of aerosols collected by two different sampling devices in a semi-arid region of Tunisia is compared by means of compositional perturbation vectors and biplots. This set of underused mathematical tools belongs to a family of statistics created specifically to deal with compositional data. The two sampling devices operate at a flow rate of about $17\,\mathrm{L\,min^{-1}}$, with a cut-off diameter of

$10\,\mu\mathrm{m}$. The first device is a low-cost laboratory-made system, where the largest particles are removed by gravitational settling in a vertical tube. This new system will be compared to the second device, a brand-new standard commercial PM₁₀ sampling head, where size segregation is achieved by particle impaction on a metal surface. A total of 44 elements (including rare earth elements or REE, together with Al, As, Ba, Be, Ca, Cd, Co, Cr, Cu, Fe, K, Li, Mg, Mn, Mo, Na, Ni, P, Pb, Rb, S, Sc, Se, Sr, Ti, Tl, U, V, Zn, and Zr), was analysed in sixteen paired samples, collected during a two-week field campaign in Tunisian

dry lands, close to source areas, with high levels of large particles. The contrasting meteorological conditions encountered during the field campaign allowed a broad range of aerosol compositions to be collected, with very different aerosol mass concentrations. No compositional differences were observed between samples collected simultaneously by the two devices. The mass concentration of the particles collected was estimated through chemical analysis, and results for the two sampling devices were also very similar to those obtained from an on-line aerosol weighing system, TEOM (tapered element oscillating

micro-balance), installed next to them. Results suggest that the commercial PM₁₀ impactor head can therefore be replaced by the decanter, without any measurable bias, for the determination of chemical composition, and for further assessment of PM₁₀ concentrations in source regions.



## 1   Introduction

At a global scale, mineral dust or mineral aerosols could represent about 40% of the total amount of particles injected into the atmosphere each year (Boucher et al., 2013; Huneeus et al., 2011). Studying atmospheric mineral dust, which modifies atmospheric radiation and alters cloud properties, thus impacting climate, is essential to better understand the evolution of Earth's climate system (e.g. Mahowald et al., 2011). Mineral dust is also an important source of nutrients necessary for phytoplankton growth in the open ocean (e.g. Okin et al., 2011) and for terrestrial plant development (e.g. Okin et al., 2004). Most of the

mineral dust present in the atmosphere comes from West Africa (Prospero and Nees, 1986; N'Tchayi Mbourou et al., 1997), with the Sahara as the main source (e.g. Ginoux et al., 2004). Accurate measurement of the chemical composition of aerosols is necessary for source tracing in aeolian studies (e.g. Scheuvens et al., 2013) which require aerosol data to assess global land degradation and climate change (e.g. Chappell et al., 2018).

In source regions of dry erodible material, high local wind speeds can move the largest and heaviest coarse soil particles

(between 50 and 200 μm in diameter) on the soil surface, while the smaller particles (less than 70 μm in diameter) move by saltation, a jumping movement near the soil surface. Collisions between these particles and aggregates of the finest particles present at the soil surface release a large spectrum of smaller particles into the air (Marticorena and Bergametti, 1995; Marticorena et al., 1997; Alfaro and Gomes, 2001). These fine particles, particularly those smaller than 10 μm in diameter ($PM_{10}$), can be transported by wind at higher altitudes over long distances (Gillette, 1981; Gomes et al., 1990; Shao et al., 1993; Shao,

2008). These particles are also a key parameter in air quality control (Kuklinska et al., 2015).

Efforts are made in atmospheric sciences to develop devices able to prevent unwanted collection of the largest particles with a 10 μm cut-off diameter. Commercially available standard sampling devices are commonly used to collect fine particles. One of the most popular is the PM10 sampling head, where size segregation is obtained by removal of the largest particles through impaction on an aluminium alloy plate. This process may however contaminate aerosol samples with metal particles,

because of friction between coarse particles and the metallic parts of the system. This is not an issue for simple aerosol mass determination, but could generate problems if the objective is to define the chemical composition of airborne particles. Among other aerosol sampling head systems is the cyclone sampling device, where particles are separated by centrifugal force. Cyclone walls may be made of glass instead of metal, thus reducing potential secondary emission effects. This system is nevertheless difficult to manage, because of its sensitivity to air pump flow rate (Haig et al., 2016). Impinger systems present a liquid

impaction surface (Yu et al., 2016), and are well adapted for bio-aerosols, but not for mineral particles. In this study, the potential of a new $PM_{10}$ sampling head is evaluated in terms of mass collection efficiency and chemical composition accuracy. It uses the decantation principle, with a very simple design and at low cost. Particle separation in this 125 mm-diameter Vertical Tube Decanter (VTD) system is based on gravitational settling counteracted by upward airflow. This system prevents collision between airborne particles and aerosol collector surfaces, so that sample contamination by metallic surface abrasion

is minimized.

The objective of this paper is to show that a low-cost decanter tube can replace an impaction-based $PM_{10}$ sampling head for proper aerosol sampling. Source regions are good places to test possible biases introduced by the sampling head device in





fine aerosol sampling, because coarse aerosols larger than 10 µm are often present. Differences in cut-off diameter tuning will
lead to differences in aerosol sample mass and chemical composition, as different amounts of the coarse particles present in
the source zone will be collected. It is for this reason that we decided to compare the performance of two different sampling
heads in a dry region of Tunisia. Aerosol chemical composition, including Rare Earth Elements (REE), and mass concentration
of aerosols were measured at the same time by two devices: the newly designed stainless steel decanter, VTD, and a brand-
new aluminium alloy commercial $PM_{10}$ (hereafter PM10), both operating at a flow rate of about $17\ \mathrm{L\,min^{-1}}$. The chosen
sampling station is part of the International Network to study Deposition and Atmospheric composition in AFrica (INDAAF),
and is equipped with a reference instrument for mass concentration measurements, a $PM_{10}$ automatic weighing device (Tapered
Element Oscillating Microbalance, TEOM). Masses deduced from elemental analysis of samples collected by each device were
compared with one another, and also with this third system, operating at the same flow rate.

## 2   Materials and methods

### 2.1   Aerosol sampling and direct measurements

Sixteen paired samples were collected during a field experiment of two weeks, at the Institut des Régions Arides campus,
20 km north of the city of Medenine, Tunisia. The collection site (33º29'58.62" N - 10º38'35.2" E), surrounded by dry lands,
is 5 km south-west of the Boughara Gulf. The two sampling devices were fixed to the roof of the highest building on campus,
about 20 m above ground level. Both VTD and PM10 were attached to a tubular stand, with a distance of about 30 cm
between them (Figure 1), to facilitate comparison of results. Aerosol samples were collected continuously from 2016/03/29 to
2016/04/07 using polysulfone open-face filter holders (Nalgene®), and mixed cellulose ester filters, with a pore size of 0.45 µm
(Whatman®). The filters were changed twice a day for each device at the same time: around 8:30 AM and 7:30 PM, except for
one pair, which was exposed for 24 hours.

    Figure 2 shows the internal structure of the commercial PM10 sampling head (Tecora, Paris, France) installed for the present
study with an aluminium alloy sampling plate. In the VTD system installed beside it, air is pumped at the top of the tube
and enters by the bottom of the tube. Fine particles are dragged upwards by the airflow and collected by the filter, but the
largest particles do not reach the filter because of their weight. The terminal settling velocity for a particle of diameter D in a
gravitational field is calculated using Stokes' law (e.g. Calvert, 1990):

$$v_g = \frac{D^2(\rho_p - \rho_{air})g}{18\mu_{air}}$$

where $v_g$ is the velocity of the particle when the steady state is reached; $\rho_p$ is particle density; $\rho_{air}$ is air density; g is gravita-
tional acceleration; and $\mu_{air}$ is the dynamic viscosity of air.

    When a particle is in the upward airflow, it is pulled up unless its gravitational settling velocity is greater than the airflow
velocity, in which case it will settle down. A cut-off point occurs when gravitational velocity is equal to air velocity: only
particles smaller than this cut-off size can reach the top of the VTD system and thus be collected on the filter. With a flow rate
of $17\ \mathrm{L\,min^{-1}}$, the Reynolds number is equal to 45 inside the VTD. A laminar flow and therefore a constant air velocity in the



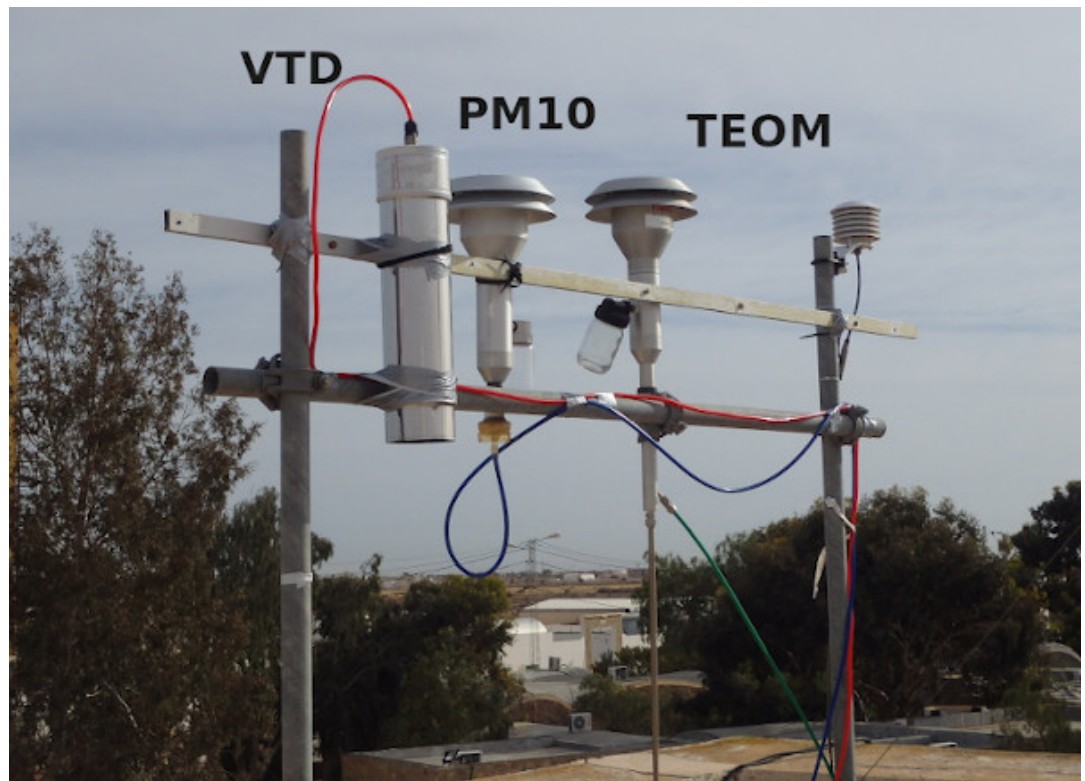

**Figure 1.** From left to right, the VTD system, the PM10 sampling head, and TEOM.

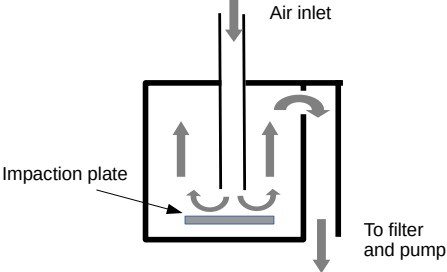

**Figure 2.** Internal structure of PM10 sampling head



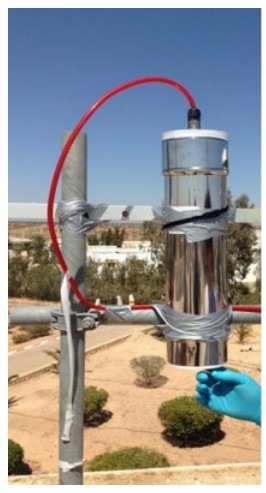
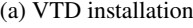

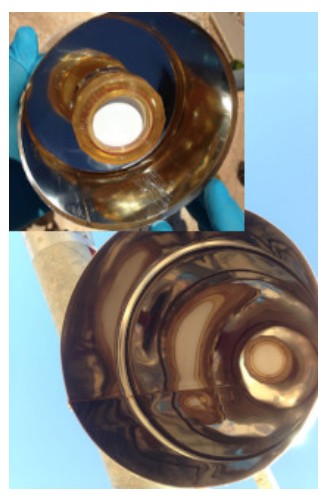

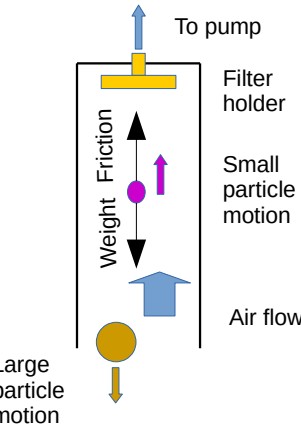

(a) VTD installation          (b) Filter inside the decanter tube          (c) Diagram of decantation system

**Figure 3.** The VTD system

tube can be assumed. The steady state settling velocity of a particle then is reached when:

$$v_g = v_{air} = \frac{F_{air}}{\pi r^2}$$

where $v_{air}$ is the upward air velocity, $F_{air}$ is the pumped air flux, and r is the radius of the cylindrical VTD system, which is about six times smaller than its height. The cut-off diameter ($D_{cut-off}$) can thus be rewritten as follows:

$$D_{cut-off} = \sqrt{\frac{18\mu_{air}F_{air}}{(\rho_P - \rho_{air})g\pi r^2}}$$

Larger particles are not collected because of their gravitational settling velocity. The $D_{cut-off}$ value varies as a function of the pumped air flux when all the other parameters are fixed (Figure 4), so that it can easily be tuned to 10 µm. In an ambient air loaded with particles larger than 10 µm, the particle mass collected by VTD should be equal to the mass collected by the commercial PM10 head, if the above-mentioned hypotheses are verified.

A Tapered Element Oscillating Microbalance (TEOM, Thermo Scientific), equipped with the same commercial PM10 head, was also installed beside the VTD and the PM10 systems (Figure 1). It measures the mass concentration of airborne particles directly, providing values considered as references for further comparison. A Portable Laser Aerosol Spectrometer (PLAS, Model 1.108/1.109, Grimm), which measures particle size distribution over a large size range, was also installed ca. 3 m away from these three systems. A 1.111 Radial symmetric sampling head (Grimm) was installed at the air inlet of the instrument to ensure reasonable capture efficiency for large particles. The PLAS measures the number of particles within 15 diameter intervals between 16 diameter channels of 0.30, 0.40, 0.50, 0.65, 0.80, 1.0, 1.6, 2.0, 3.0, 4.0, 5.0, 7.5, 10, 15, 20 and 25 $\mu$m. Counting by PLAS is converted into mass assuming lognormal distribution of spherical particles. The volume of $N_i$ particles in the $[d_i, d-i+1]$ diameter interval is equal to $V_i = N_i \frac{\pi \overline{d}^3}{6}$, where $\overline{d}$ is the geometric mean of $d_i$ and $_{di+1}$. With a particle



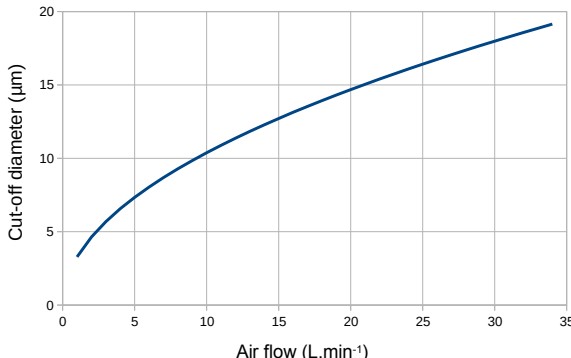

**Figure 4.** particle cut-off diameter (μm), in a cylinder system with a diameter of 125 mm, as a function of airflow

density $\rho$, commonly chosen to equal $2.2\ \mathrm{g\,cm^{-3}}$, the PM$_{10}$ mass $m_{10}$ is equal to the sum of all the channels under 10 μm:

$$m_{10} = \sum \rho V_i$$

while the mass of coarse particles larger than 10 μm is obtained by summing the channels over 10 μm. This coarse particle
mass fraction should not be sampled by our sampling devices.

## 2.2 Washing procedure for sampling instruments

Prior to the field experiment, in the laboratory, 50 Petri dishes (PALL, filter storage box) were washed with detergent, and
rinsed with tap water, after which they were soaked in osmosed water containing 2% of Decon90® for at least 15 hours. They
were then thoroughly rinsed with tap water followed by osmosed water, before being soaked in acidified (HCl 1%) osmosed
water for 3 days. Finally, the Petri dishes were rinsed with MilliQ®water ($18\ \mathrm{M\Omega\,cm^{-1}}$) and dried in an ISO-2 laminar flow
hood. The filter holders and their PP boxes were cleaned using the same procedure. The PM10 head was disassembled and each
part was washed with tap water and detergent, and then soaked in osmosed water containing Decon90®for several minutes.
Finally, each part was washed with osmosed and MilliQ water ($18\ \mathrm{M\Omega\,cm^{-1}}$), and dried in the laminar flow hood. The tube of
the VTD was washed with detergent, and rinsed with Milli-Q water, before also being dried in the laminar flow hood.

## 2.3 Sample digestion

The filters coated with dust samples were brought back to the laboratory (ISO-7 clean room) and digested in sealed Teflon®
(PTFE) digestion vessels by $3\ \mathrm{mL}$ of a mixture of sub-boiled HNO$_3$/HF (9:1) for 18 h on a heater plate at 125°C. All the
Teflon vessels were previously cleaned with the detergent/acid procedure described above, completed with blank digestion.
At the end of digestion, each vessel was opened, and the temperature of the heater plate was raised to 135°C, until complete
evaporation of all liquid. The temperature of the heater plate was then lowered to 80°C, and $3\ \mathrm{mL}$ of a 30% nitric acid solution
was added to each vessel, which was then sealed. Two hours later, the content of each vessel was transferred into a 60 mL





polypropylene bottle (thoroughly detergent/acid cleaned), by adding Milli-Q water. Laboratory blanks (no filter), 4 field blanks (pristine filter), and 2 finely ground geostandards (SCO-1 and MAG-1 from USGS) were also prepared following the same digestion procedure.

## 2.4 Chemical analyses

An ARCOS (Spectro-Ametek) ICP-AES, equipped with a CETAC ultrasonic nebuliser, was used for elemental determination of Al, Ba, Ca, Cr, Fe, K, Li, Mg, Mn, Na, P, Sc, S, Sr, Ti, Zn, and Zr. A Field-Sector High-Resolution Inductively Coupled Plasma-Mass-Spectrometer (FS-HR-ICP-MS) Thermo Element 2, equipped with a concentric micro-nebuliser in a cyclonic nebulisation chamber was used for elemental determination of As, Be, Cd, Co, Cu, Mo, Ni, Pb, Rb, Se, Tl, U, V, and REE. External linear calibration was performed for all elements analysed with ICP-AES, by measuring a set of multi-elementary solutions with concentrations up to $250\,\mu g\,L^{-1}$. Zero was computed as the average of eight replicates of a blank sample (ultra-pure nitric acid diluted in Mili-Q water). High-resolution analysis avoids polyatomic interference for elements lighter than arsenic, and also for REE (Heimburger et al 2013). The FS-HR-ICP-MS was externally calibrated for all elements analysed, with fourteen replicates of a blank solution and five replicates of a $1\,\mu g\,L^{-1}$ multi-elementary solution. The first analytical detection limit was obtained with analytical blanks, and digestion with dilution water and acid reagents only, while the second field detection limit was obtained with blank filters transported to the field. For most of the elements, quantities found in blank filters were higher than analytical detection limits, so that blank correction used the average quantity found in blank filters. For a few elements (Pr, Eu, Tb, Dy, Ho, Tm, and Lu), blanks were below detection limits so no blank correction was made. Seven elements (As, Cd, Cr, Mo, Ni, Sc, and Se) are not discussed, because at least one measured value was below the field or analytical detection limit and, therefore, cannot be handled by the statistical tools used here.

## 2.5 Validation of analytical methods

There is no commercially available certified reference material comparable to the fine aerosols collected on filters. Two geo-standards were therefore used as proxies: SCO–1 (typical of Upper Cretaceous silty marine shale), and MAG–1 (a fine-grained grey-brown clayey mud with low carbonate content, from the Wilkinson Basin of the Gulf of Maine). They were hand-crushed for 30 min in an agate mortar to approximate aerosol grain size. The powders produced were deposited on a filter at the smallest amount that can be weighed, around 10 mg with an accuracy of 0.2 mg, to obtain a mass as close as possible to field aerosol samples. A table with individual recovery rates, as well as individual measurement results for each certified element and aerosol sample is proposed as supplementary material (Table B2). Recovery rates for most elements ranged from 80% to 120% for SCO–1 and MAG–1, but could not be calculated for S, Se, and Tm, because no value was available for comparison.

## 2.6 Computation of total aerosol mass concentration

The PM$_{10}$ mass concentration was not directly measured because of the low expected weight and the nature of the filter. In this region, almost all the particle mass can be assumed to be carried by silicate crustal particles, sea-salts, sulphuric acid ($H_2SO_4$),





and additional calcium in the form of calcium carbonate ($CaCO_3$). A chemical reaction occurs between calcium carbonate and sulphuric acid, producing gypsum ($CaSO_4.2H_2O$), and preventing the simultaneous presence of sulphuric acid and calcium carbonate (Mori et al., 1998). If carbonate predominates over sulphuric acid, the total particle mass concentration is computed as:

$$[particles]_{air} = [crust\ particles]_{air} + [sea\ salt]_{air} + [CaSO_4 \cdot 2H_2O]_{air} + [CaCO_3]_{air}$$

If sulphuric acid predominates over carbonate, then:

$$[particles]_{air} = [crust\ particles]_{air} + [sea\ salt]_{air} + [CaSO_4 \cdot 2H_2O]_{air} + [H_2SO_4]_{air}$$

$[particles]_{air}$ is estimated using aluminium, and a crustal composition model where aluminium accounts for 7.01% of the mass (Bowen, 1966):

$$[crust\ particles]_{air} = \frac{[Al]_{air}}{(X_{Al})_{crust\ model}} = \frac{[Al]_{air}}{7.01\%}$$

$[sea\ salt]$ is estimated using sea-salt sodium, and a seawater composition model (Dickson and Goyet, 1994) where sodium accounts for 30.9% of sea-salt mass. Sea-salt sodium is deduced by subtracting crustal sodium from total sodium, crustal sodium being deduced from aluminium (Rahn, 1976), and a crustal composition model where the Na/Al ratio is equal to 0.0887 (Bowen, 1966):

$$[Na_{crustal}]_{air} = [Al]_{air}\left(\frac{[Na]}{[Al]}\right)_{crust\ model} = [Al]_{air} \cdot 0.0887$$

$$[Na_{sea\ salt}]_{air} = [Na_{total}]_{air} - [Na_{crustal}]_{air}$$

$$[sea\ salt]_{air} = \frac{[Na_{sea\ salt}]_{air}}{(X_{Na})_{seawater\ model}} = \frac{[Na_{sea\ salt}]_{air}}{30.9\%}$$

$[CaSO_4 \cdot 2H_2O]$, $[CaCO_3]$ and $[H_2SO_4]$ are calculated using additional calcium and additional sulphur not included in crustal and sea-salt estimation. Ca* and S* are defined respectively as calcium and sulphur of neither sea-salt nor crustal origin. Ca* and S* are computed using the same crustal and sea-salt composition models previously used:

$$[Ca^*]_{air} = [Ca]_{air} - [Na_{sea\ salt}]_{air} \cdot \left(\frac{[Ca]}{[Na]}\right)_{sea\ salt\ model} - [Al]_{air} \cdot \left(\frac{[Ca]}{[Al]}\right)_{crustal\ model}$$

$$[Ca^*]_{air} = [Ca]_{air} - [Na_{sea\ salt}]_{air} \cdot 0.037 \cdot [Al]_{air} \cdot 0.193$$

$$[S^*]_{air} = [S]_{air} - [Na_{sea\ salt}]_{air} \cdot \left(\frac{[S]}{[Na]}\right)_{sea\ salt\ model} - [Al]_{air} \cdot \left(\frac{[S]}{[Al]}\right)_{crustal\ model}$$



$$[S^*]_{air} = [S]_{air} - [Na_{sea\ salt}]_{air} \cdot 0.0843 \cdot [Al]_{air} \cdot 0.0099$$

Depending on the resulting products of calcium carbonate with sulphuric acid reaction, the mass associated to additional calcium and sulphur is computed as follows:

$$[CaSO_4 \cdot 2H_2O]_{air} + [CaCO_3]_{air} = [S^*]_{air} \frac{M_{CaSO_4 \cdot 2H_2O}}{M_S} + \left([Ca^*]_{air} - [S^*]_{air} \frac{M_{Ca}}{M_S}\right) \frac{M_{CaCO_3}}{M_{Ca}}$$

$$[CaSO_4 \cdot 2H_2O]_{air} + [H_2SO_4]_{air} = [CaS^*]_{air} \frac{M_{CaSO_4 \cdot 2H_2O}}{M_{Ca}} + \left([S^*]_{air} - [Ca^*]_{air} \frac{M_S}{M_{Ca}}\right) \frac{M_{H_2SO_4}}{M_S}$$

where $M_X$ is the molar mass of the compound or element X.

## 2.7 Multivariate analysis for compositional data

Compositional data are, by nature, difficult to handle straightforwardly. Any given component cannot vary independently from the others, because the sum of all components, equal to 100%, includes components that are not measured. If this closure constraint is not taken into account, spurious correlations and biased conclusions are to be expected (Van der Weijden, 2002). Appropriate mathematical tools must therefore be selected to overcome this drawback. These questions are extensively discussed in Aitchison (1986, 1992, 2005); Barceló-Vidal et al. (2001); Filzmoser et al. (2009); Egozcue, J.J. et al. (2003). Briefly, the suitable sample space of any compositional vector $\mathbf{x}$, representing a D–part subset of a whole $\mathbf{x} = [x_1, \ldots, x_D]$, is the simplex $S^D$, as defined by Aitchison (1986). This technique is particularly well adapted to situations where elemental ratios are more relevant than absolute concentrations.

Let $\mathbf{x} = [x_1, \ldots, x_D]$ and $\mathbf{y} = [y_1, \ldots, y_D]$ denote two compositional vectors in $S^D$. Then $\mathbf{z}$, corresponding to the perturbation of $\mathbf{x}$ by $\mathbf{y}$, in $S^D$ is given by:

$$\mathbf{z} = \mathbf{x} \oplus \mathbf{y} = C\,[x_1 y_1, \ldots, x_D y_D]$$

With C the closure-to-unity operation defined as:

$$C(\mathbf{x}) = \left[ \frac{x_1}{\sum\limits_{i=1}^{D} x_i}, \ldots, \frac{x_D}{\sum\limits_{i=1}^{D} x_i} \right]$$

The neutral element of the perturbation is $e = C[1, \ldots, 1] = \left[\frac{1}{D}, \ldots, \frac{1}{D}\right]$, and $x = x \oplus e$ while the perturbation vector expression compositional change from y to x, noted $\mathbf{x} \ominus \mathbf{y}$ is equal to $\mathbf{x} \oplus \mathbf{y}^{-1}$ with $\mathbf{y}^{-1} = C\left[y_1^{-1}, \ldots, y_D^{-1}\right]$ (von Eynatten et al., 2002; Aitchison and Ng, 2005). The centred log-ratio (*clr*) transformation is commonly performed to open the data before applying any multivariate techniques based on correlation:

$$clr(x) = \left[ ln\frac{x_1}{gm(x)}, \ldots, ln\frac{x_D}{gm(x)} \right]$$





where $gm(x)$ denotes the geometric mean of the D parts: $gm(x) = \left( \prod_{i=1}^{D} x_i \right)^{\frac{1}{D}}$. A Principal Component Analysis (PCA) can then be computed on transformed data to summarize the structure of the data in a lower dimensional space (ideally two). A compositional biplot, where both samples and variables are plotted in the same space can be used as a user-friendly graphical

representation, but it differs from the original biplot by Gabriel (1971) in the sense that rays formed by the variables are proportional to the standard deviation of their log-ratios, and that the length of a link between arrow heads of two rays represents the standard deviation of the log-ratio between these compositional parts (Suárez et al., 2016). Practically, the "acomp" and "princomp" functions used here were provided by the "compositions" package for the R software (R Core Team, 2014) which was specifically designed to analyse compositional data (van den Boogaart et al., 2014).

## 3 Result and discussion

### 3.1 Variability of sampling conditions

The sampling site can be influenced by local and remote soil dust emission, by sea salt, and by anthropogenic emissions. During the sampling campaign, a broad variety of meteorological conditions was observed, allowing different aerosol sources to be sampled. Average local wind speed varied from about 1 to 7 $\mathrm{m\,s^{-1}}$ with no preferred direction (appendix, Figure C1).

Backward air trajectories are presented for each sample pair in appendix C, indicating their differences in origin, leading to a variety of conditions for aerosol loading. Atmospheric aerosol loading presented a large range of values, from 5 to 26 $\mathrm{\mu g\,m^{-3}}$ for sea salt and from 9 to 730 $\mathrm{\mu g\,m^{-3}}$ for crustal dust (Table 1), with great variation between marine versus crustal proportions in any given sample pair.

### 3.2 Size distribution of the sampled aerosol

The fraction of particles larger than 10 μm suspended in the air is shown by PLAS measurements. For the entire field experiment, this coarse fraction represents on average 34% of the total mass concentration of aerosols, as plotted for a given day in Figure 5. If the cut-off diameter was not the same for each sampling head in a given sample pair, the amount of large particles collected would produce differences in sampled aerosol mass and chemical composition.

### 3.3 Total aerosol mass concentration in air

Comparisons of the collected masses between VTD, PM10, and the reference instrument TEOM are shown in Figure 6 and Table 1. Plotted concentrations vary from 21 to 680 $\mathrm{\mu g\,m^{-3}}$ within a range which can be cleverly plotted using a square root scale (Verrall and Bell, 1969). Masses of particles collected by VDT and PM10, deduced from calculations using Al, Na, S, and Ca, fit the TEOM values (Figure 6a). Similar results are observed for each VTD and PM10 sample pair (Figure 6b), indicating the same collection efficiency for both sampling heads, and hence the same cut-off diameter. The median value

of the relative mass differences between VTD and PM10 is +12%, and values range from -3% to +22%. To summarize, the differences observed between aerosol masses measured by the three sampling systems are much lower than the daily variability





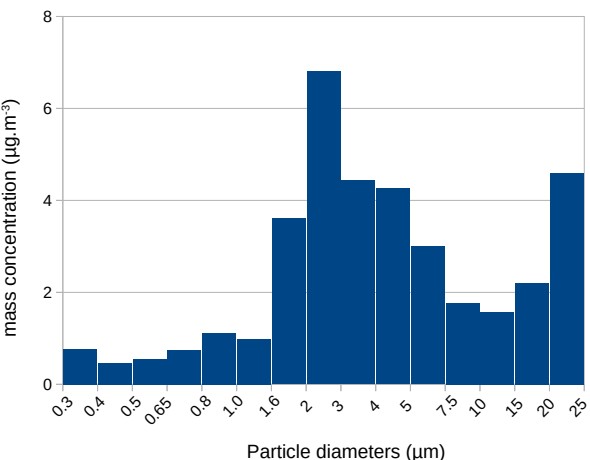

**Figure 5.** Daily average particle mass concentration size distribution in air on April 6. Measured using the Grimm PLAS

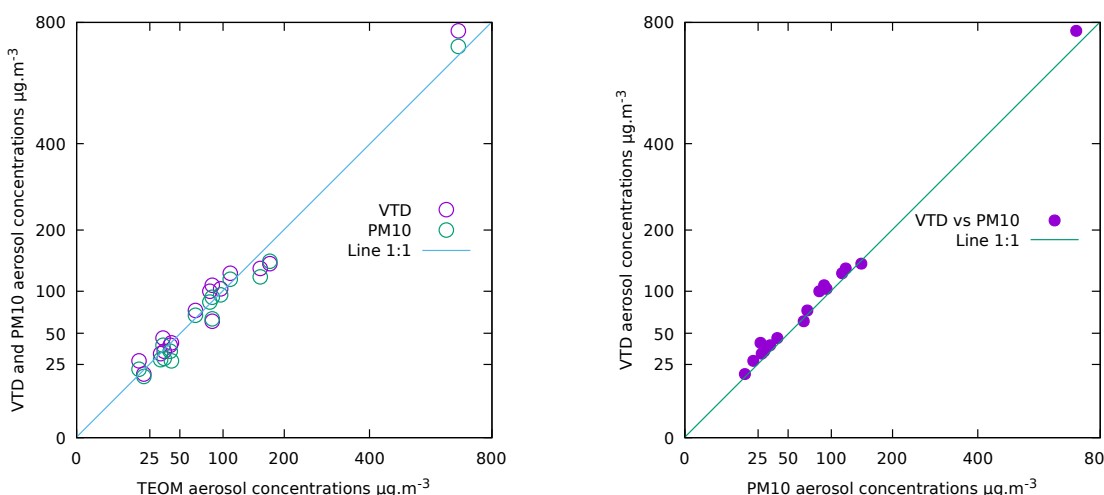

(a) Plot of chemically deduced mass of VTD and PM10 sampling heads versus TEOM measurement.

(b) Plot of chemically deduced mass of VTD versus PM10 sampling heads.

**Figure 6.** Comparisons of sample masses using square-root scale. The lines x=y are also shown

observed during the field experiment. The coherence between direct measurement of masses (TEOM) and "chemical" weighing shows that substances not taken into account in our chemical budget (ammonium and organic molecules) do not significantly contribute to the total aerosol mass here.





| Sample name | Start | End | [Sea salt] $\mu g\,m^{-3}$ | [Crustal] $\mu g\,m^{-3}$ | [Calcium species] $\mu g\,m^{-3}$ | [Total VTD] $\mu g\,m^{-3}$ | [Total PM10] $\mu g\,m^{-3}$ | TEOM $\mu g\,m^{-3}$ |
|---|---|---|---|---|---|---|---|---|
| YX03/04 | 29/03/2016 18:40 | 30/03/2016 09:12 | 2.6 ± 0.8 | 8.8 ± 0.3 | 7.6 ± 0.4 | 19 ± 1 | 17 ± 2 | 21 |
| YX05/06 | 30/03/2016 09:47 | 30/03/2016 18:26 | 7 ± 1 | 10 ± 0.3 | 10.4 ± 0.7 | 27 ± 2 | 22 ± 3 | 18 |
| YX07/08 | 30/03/2016 19:08 | 31/03/2016 09:03 | 5 ± 1 | 20.1 ± 0.6 | 7 ± 2 | 33 ± 2 | 28 ± 2 | 33 |
| YX09/10 | 31/03/2016 09:38 | 31/03/2016 18:29 | 5 ± 1 | 45 ± 1 | 12 ± 3 | 62 ± 4 | 66 ± 2 | 86 |
| YX11/12 | 31/03/2016 19:01 | 01/04/2016 09:09 | 9 ± 1 | 21.8 ± 0.7 | 8.7 ± 0.6 | 40 ± 1 | 34 ± 2 | 41 |
| YX13/14 | 01/04/2016 09:39 | 01/04/2016 18:19 | 20 ± 2 | 105 ± 3 | 14 ± 1 | 140 ± 4 | 145 ± 4 | 175 |
| YX15/16 | 01/04/2016 18:49 | 02/04/2016 09:38 | 5.1 ± 0.9 | 115 ± 3 | 4.6 ± 0.6 | 124 ± 4 | 116 ± 3 | 111 |
| YX17/18 | 02/04/2016 10:08 | 02/04/2016 19:23 | 7 ± 2 | 730 ± 20 | 32 ± 2 | 769 ± 20 | 711 ± 20 | 679 |
| YX19/20 | 02/04/2016 19:49 | 03/04/2016 09:52 | 17 ± 1 | 68 ± 2 | 13.2 ± 0.6 | 99 ± 3 | 84 ± 2 | 82 |
| YX21/22 | 03/04/2016 10:19 | 03/04/2016 18:29 | 21 ± 2 | 37 ± 1 | 16.2 ± 0.9 | 75 ± 3 | 70 ± 2 | 66 |
| YX23/24 | 03/04/2016 18:59 | 04/04/2016 10:01 | 23 ± 1 | 9 ± 0.3 | 9.6 ± 0.5 | 41 ± 2 | 35 ± 1 | 42 |
| YX25/26 | 04/04/2016 10:31 | 04/04/2016 19:31 | 18 ± 2 | 71 ± 2 | 14.4 ± 0.8 | 103 ± 3 | 94 ± 3 | 97 |
| YX27/28 | 04/04/2016 20:01 | 05/04/2016 10:00 | 11 ± 1 | 15 ± 0.5 | 8.2 ± 0.5 | 35 ± 1 | 29 ± 1 | 36 |
| YX29/30 | 05/04/2016 10:24 | 06/04/2016 09:06 | 8.6 ± 0.8 | 115 ± 3 | 9.8 ± 0.4 | 133 ± 4 | 120 ± 3 | 157 |
| YX31/32 | 06/04/2016 10:33 | 06/04/2016 18:51 | 26 ± 2 | 67 ± 2 | 14.9 ± 0.9 | 108 ± 3 | 91 ± 3 | 85 |
| YX33/34 | 06/04/2016 19:19 | 07/04/2016 07:16 | 18 ± 1 | 19.3 ± 0.6 | 8.2 ± 0.5 | 45 ± 2 | 40 ± 1 | 35 |

**Table 1.** Sampling dates (local time) and dust mass calculated from chemical analysis of samples collected by VTD and PM10, and directly measured by TEOM. Grey columns give details of the three components of the total mass of samples collected with the VTD.

## 3.4 Compositional data

The aim is now to compare chemical compositions of samples collected simultaneously by both VDT and PM10, as differences may appear due to contamination, size segregation of particles, and mineralogical fractionation during sampling. Note that major and trace elements are treated separately from the REE in the following, because of the particular importance of REE as tracers of mineral particle origin (Wang et al., 2017).

### 3.4.1 Major and trace elements

The first two axes of the compositional biplot built from major and trace elements, except REE, explain 77% of the total variance (61% and 16% respectively), a high value, considering that 23 variables are taken into account for the analysis (Figure 7a). The variability between each pair of samples (i.e. collections by PM10 and VTD on the same day) appears to be much lower than the variability observed within the entire set of samples. In other words, each dust event can be characterised properly with respect to the others, independently of the sampling device used. This finding is in good agreement with a close examination of compositional changes between PM10 and VTD for each pair of samples, expressed as perturbation vectors: $VTP \ominus PM10$, with $VTP \ominus PM10 \in S^{23}$ (Figure 7b). Interestingly, the neutral element $\mathbf{e} = \left[\frac{1}{23}, \ldots, \frac{1}{23}\right] = [0.043, \ldots, 0.043]$, which indicates no perturbation is included inside all the box plot quartiles. No systematic compositional shift, in terms of





elemental ratios, can therefore be observed between the two sampling heads, at least for these elements. Note, however, that

Zn exhibits the greatest variability, suggesting noticeable random contamination.

### 3.4.2    Rare Earth Elements (REE)

In the compositional biplot built from REE, only 51% of the total variance is explained by the first two axes (Figure 8a). This value is much lower than that obtained above for the other chemical elements (77%), but with only half the number of variables. The corresponding perturbation vector diagram again shows no systematic difference between the two sampling heads ( 8b).

Because REE essentially come from a stable crustal source, log-ratios between these elements vary little within the sample set (almost ten times less than the variability observed for the other elements). This stability explains why the percentage of variance expressed by the first two principal components is so low.

To test if the differences observed between the two systems might be explained solely by analytical error, the behaviour of identical duplicate samples was simulated: 16 new pairs of compositions were generated, by pairing each VTD sample with a

modified sample where each REE measurement was randomly shifted inside the given uncertainty interval of that REE. These new pairs of simulated samples were then represented as a biplot (appendix Figure D1), producing results very similar to those observed for the real (VTD and PM10) paired sample.

In the source region investigated, where particle mass concentrations ranged from 18 to 680 $\mu g\,m^{-3}$ according to TEOM values, the chemical composition of the $PM_{10}$ aerosol fraction was therefore unaffected by the sampling head design.





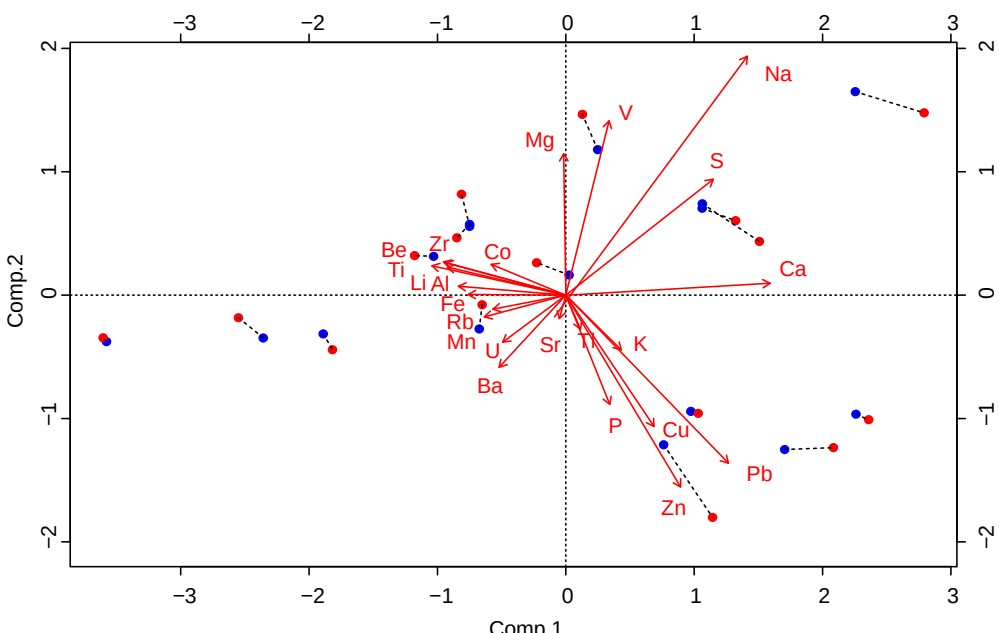

(a) Biplot for the two sampling devices, red PM10, blue VTD. Lines between red and blue points link paired samples. Percentages of variability explained by the first two components are 61% and 16%, a total of 77%.

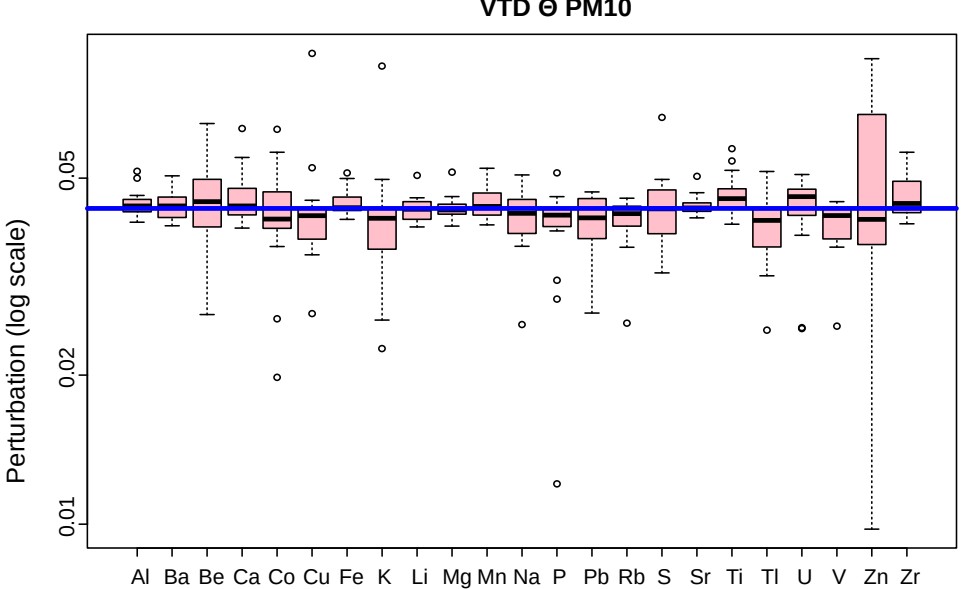

(b) Perturbation diagram as box plots (median, quartiles, likeliness distance at the end of dotted segments, and outliers as circles) for paired samples, measured by PM10 and VTD, for all elements except REE. The horizontal blue line represents no perturbation.

**Figure 7.** All elements except REE





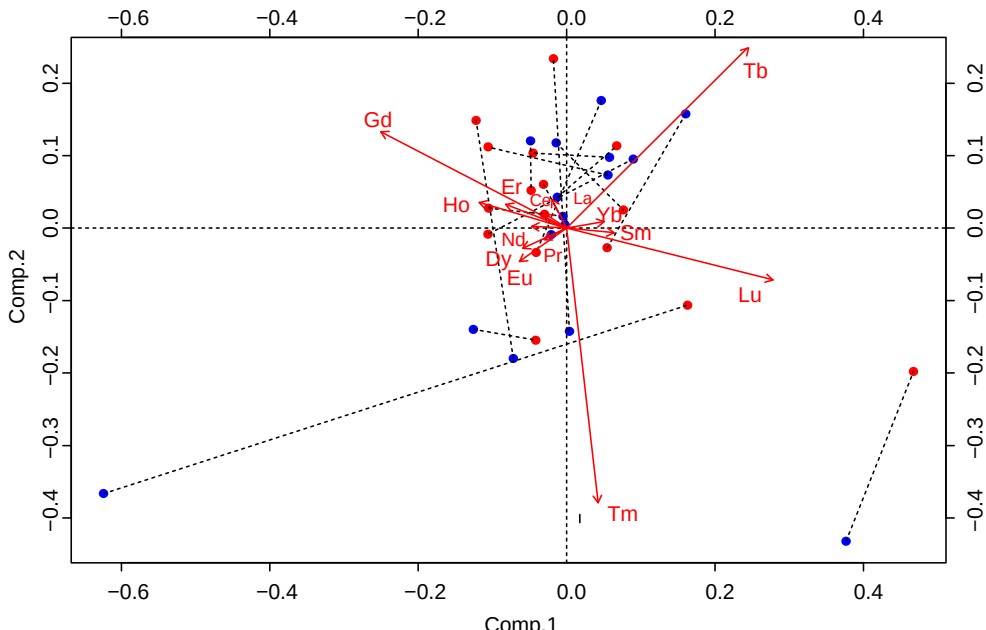

(a) REE biplot for the two sampling devices (red PM10, blue VTD). Lines between red and blue points link paired samples. Percentages of variability explained by the first two components are 29% and 22%, a total of 51%.

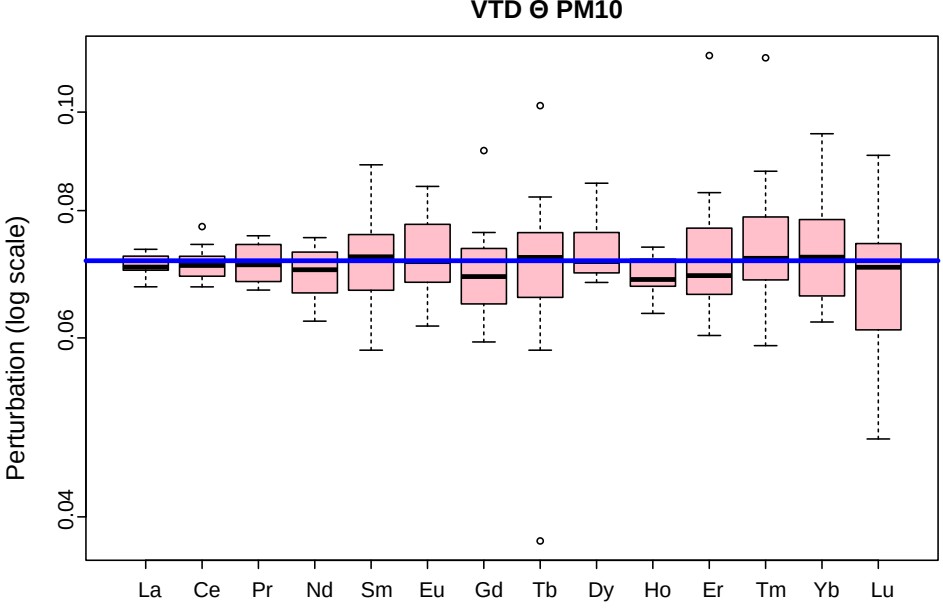

(b) REE perturbation diagram as box plots (median, quartiles, likeliness distance at the end of dotted segments, and outliers as circles) for paired samples, measured by PM10 and VTD. The horizontal blue line represents no perturbation.

**Figure 8.** REE



## 4 Conclusions

Compositional Data Analysis tools can be used to present large sets of measurements at a glance, allowing us to perceive the compositional similarity of paired samples quickly and directly. No significant differences between the laboratory-made decanter sampling head and the commercial $PM_{10}$ sampling head (based on impaction) were observed in terms of aerosol composition (including REE) and total mass concentration, for samples collected in a source region, under very different meteorological conditions. Consequently, both devices can be used for the determination of mass and chemical composition of aerosols in source regions, or even simply to determine mass by gravimetry. An aerosol survey network can therefore be built using a combination of the two sampling devices without any measurable consequences on data reliability or consistency. This would also be the case for a time series if a PM10 was replaced by a VTD, or vice versa. The new $PM_{10}$ sampling head system, based on a decanter, is simple to build, at low cost, and can be made available anywhere in the world.





## Appendix A: Air concentrations, measured values

Raw data of the paper are presented in Tables A1, A2 and A3 for ICP-AES measurements, in Tables A4 and A4 for ICP-MS measurements, and in Tables A6 and A7 for REE with ICP-MS.



**Table A1.** Elemental air concentrations measured with ICP-AES

| Element | | Al | Ca | Fe | K | Mg | Na |
|---|---|---|---|---|---|---|---|
| Wavelenght (nm) | | 396.2 | 396.847 | 238.2 | 766.491 | 279.553 | 589 |
| Analytical DL (ng) | | 0.02 | 0.005 | 0.1 | 0.2 | 0.003 | 1 |
| Field DL (ng) | | 0.5 | 13 | 0.8 | 3 | 1 | 79 |
| Sample name | Air volume ($m^3$) | $\mu g \, m^{-3}$ | $\mu g \, m^{-3}$ | $\mu g \, m^{-3}$ | $\mu g \, m^{-3}$ | $\mu g \, m^{-3}$ | $\mu g \, m^{-3}$ |
| YX03 (VTD) | 10.52 | 0.63 ± 0.02 | 1.7 ± 0.1 | 0.4 ± 0.01 | 0.86 ± 0.04 | 0.32 ± 0.01 | 0.9 ± 0.3 |
| YX04 (PM10) | 12.59 | 0.55 ± 0.02 | 1.6 ± 0.1 | 0.34 ± 0.01 | 1.46 ± 0.05 | 0.29 ± 0.01 | 1.3 ± 0.2 |
| YX05 (VTD) | 5.89 | 0.71 ± 0.02 | 2.6 ± 0.1 | 0.44 ± 0.02 | 2.05 ± 0.08 | 0.33 ± 0.02 | 2.2 ± 0.5 |
| YX06 (PM10) | 6.68 | 0.58 ± 0.02 | 2.2 ± 0.1 | 0.34 ± 0.02 | 2.74 ± 0.1 | 0.27 ± 0.01 | 1.8 ± 0.4 |
| YX07 (VTD) | 10.02 | 1.43 ± 0.04 | 2.5 ± 0.1 | 0.87 ± 0.03 | 1.77 ± 0.06 | 0.65 ± 0.02 | 1.7 ± 0.3 |
| YX08 (PM10) | 10.8 | 1.19 ± 0.04 | 2.1 ± 0.1 | 0.73 ± 0.02 | 0.76 ± 0.03 | 0.54 ± 0.02 | 1.2 ± 0.3 |
| YX09 (VTD) | 6.04 | 3.2 ± 0.1 | 3.7 ± 0.2 | 1.78 ± 0.06 | 1.58 ± 0.07 | 1.41 ± 0.05 | 1.8 ± 0.5 |
| YX10 (PM10) | 6.77 | 3.3 ± 0.1 | 3.5 ± 0.2 | 1.81 ± 0.06 | 1.72 ± 0.07 | 1.5 ± 0.05 | 2.3 ± 0.4 |
| YX11 (VTD) | 9.59 | 1.54 ± 0.05 | 2.4 ± 0.1 | 0.94 ± 0.03 | 0.8 ± 0.04 | 0.98 ± 0.03 | 2.9 ± 0.3 |
| YX12 (PM10) | 10.92 | 1.31 ± 0.04 | 2.1 ± 0.1 | 0.81 ± 0.03 | 0.76 ± 0.03 | 0.85 ± 0.03 | 2.7 ± 0.3 |
| YX13 (VTD) | 5.71 | 7.5 ± 0.2 | 4.2 ± 0.2 | 4.1 ± 0.1 | 9.32 ± 0.3 | 3.3 ± 0.1 | 6.9 ± 0.6 |
| YX14 (PM10) | 6.64 | 7.7 ± 0.2 | 3.5 ± 0.2 | 4.3 ± 0.1 | 13.7 ± 0.4 | 3.4 ± 0.1 | 7.6 ± 0.6 |
| YX15 (VTD) | 10.85 | 8.1 ± 0.2 | 2.2 ± 0.1 | 4.4 ± 0.1 | 2.27 ± 0.08 | 2.08 ± 0.07 | 2.3 ± 0.3 |
| YX16 (PM10) | 11.75 | 7.6 ± 0.2 | 2 ± 0.1 | 4.1 ± 0.1 | 2.05 ± 0.07 | 1.9 ± 0.06 | 2.1 ± 0.3 |
| YX17 (VTD) | 6.53 | 52 ± 2 | 3.4 ± 0.2 | 29.6 ± 0.9 | 13.8 ± 0.4 | 14.1 ± 0.4 | 6.7 ± 0.6 |
| YX18 (PM10) | 7.12 | 48 ± 1 | 3.2 ± 0.1 | 27.4 ± 0.8 | 12.7 ± 0.4 | 12.6 ± 0.4 | 6.1 ± 0.5 |
| YX19 (VTD) | 10.33 | 4.9 ± 0.1 | 3 ± 0.1 | 2.61 ± 0.08 | 2.5 ± 0.09 | 1.78 ± 0.06 | 5.8 ± 0.4 |
| YX20 (PM10) | 10.96 | 4.2 ± 0.1 | 2.2 ± 0.1 | 2.29 ± 0.07 | 1.98 ± 0.07 | 1.59 ± 0.05 | 4.4 ± 0.3 |
| YX21 (VTD) | 5.48 | 2.63 ± 0.08 | 3.7 ± 0.2 | 1.48 ± 0.05 | 1.03 ± 0.06 | 1.58 ± 0.05 | 6.9 ± 0.6 |
| YX22 (PM10) | 6.08 | 2.36 ± 0.07 | 3.5 ± 0.2 | 1.35 ± 0.04 | 1.12 ± 0.06 | 1.43 ± 0.05 | 6.9 ± 0.6 |
| YX23 (VTD) | 10.76 | 0.64 ± 0.02 | 1.9 ± 0.1 | 0.39 ± 0.02 | 1.41 ± 0.05 | 1.14 ± 0.04 | 7.1 ± 0.4 |
| YX24 (PM10) | 11.95 | 0.44 ± 0.01 | 1.17 ± 0.06 | 0.27 ± 0.01 | 1.44 ± 0.05 | 0.98 ± 0.03 | 6.4 ± 0.4 |
| YX25 (VTD) | 6.79 | 5.1 ± 0.2 | 3.4 ± 0.2 | 2.97 ± 0.09 | 1.93 ± 0.07 | 2.41 ± 0.08 | 5.9 ± 0.5 |
| YX26 (PM10) | 7.65 | 4.6 ± 0.1 | 3.2 ± 0.1 | 2.68 ± 0.08 | 1.67 ± 0.07 | 2.17 ± 0.07 | 5.8 ± 0.5 |
| YX27 (VTD) | 10.73 | 1.06 ± 0.03 | 1.98 ± 0.09 | 0.66 ± 0.02 | 0.53 ± 0.03 | 0.85 ± 0.03 | 3.6 ± 0.3 |
| YX28 (PM10) | 11.8 | 0.83 ± 0.03 | 1.64 ± 0.08 | 0.49 ± 0.02 | 0.47 ± 0.03 | 0.67 ± 0.02 | 3.1 ± 0.3 |
| YX29 (VTD) | 16.66 | 8.1 ± 0.2 | 1.37 ± 0.06 | 4.87 ± 0.15 | 4.6 ± 0.1 | 3.88 ± 0.12 | 3.4 ± 0.2 |
| YX30 (PM10) | 19.25 | 7.3 ± 0.2 | 1.2 ± 0.05 | 4.43 ± 0.13 | 5.2 ± 0.2 | 3.42 ± 0.1 | 3 ± 0.2 |
| YX31 (VTD) | 5.92 | 4.8 ± 0.1 | 4.2 ± 0.2 | 2.61 ± 0.08 | 3 ± 0.1 | 2.64 ± 0.09 | 8.5 ± 0.7 |
| YX32 (PM10) | 7.49 | 4.1 ± 0.1 | 3.2 ± 0.1 | 2.26 ± 0.07 | 2.17 ± 0.08 | 2.27 ± 0.07 | 6.7 ± 0.5 |
| YX33 (VTD) | 8.31 | 1.37 ± 0.04 | 2.2 ± 0.1 | 0.82 ± 0.03 | 0.69 ± 0.04 | 1.16 ± 0.04 | 5.7 ± 0.5 |
| YX34 (PM10) | 9.15 | 1.11 ± 0.03 | 1.8 ± 0.1 | 0.65 ± 0.02 | 0.61 ± 0.03 | 1.01 ± 0.03 | 4.9 ± 0.4 |



**Table A2.** Elemental air concentrations measured with ICP-AES, continued

| Element | Ba | Li | Mn | P | S | Sc | Sr |
|---|---|---|---|---|---|---|---|
| Wavelenght (nm) | 233.527 | 670.78 | 257.611 | 177.495 | 182.034 | 335.373 | 460.733 |
| Analytical DL (ng) | 0.001 | 0.0002 | 0.001 | 0.01 | 0.9 | 0.001 | 0.002 |
| Field DL (ng) | 0.02 | 0.002 | 0.1 | 0.2 | 85 | — | 0.05 |
| Sample name | $\mathrm{ng\,m^{-3}}$ | $\mathrm{ng\,m^{-3}}$ | $\mathrm{ng\,m^{-3}}$ | $\mathrm{ng\,m^{-3}}$ | $\mathrm{ng\,m^{-3}}$ | $\mathrm{ng\,m^{-3}}$ | $\mathrm{ng\,m^{-3}}$ |
| YX03 | $6.2 \pm 0.3$ | $0.53 \pm 0.03$ | $7.3 \pm 0.4$ | $56 \pm 2$ | $1.6 \pm 0.3$ | $0.2 \pm 0.1$ | $10.8 \pm 0.5$ |
| YX04 | $5.4 \pm 0.2$ | $0.47 \pm 0.02$ | $6.5 \pm 0.4$ | $49 \pm 2$ | $0.9 \pm 0.2$ | $<$ | $9.4 \pm 0.4$ |
| YX05 | $7.7 \pm 0.4$ | $0.54 \pm 0.04$ | $8.2 \pm 0.6$ | $74 \pm 4$ | $2.2 \pm 0.5$ | $0.2 \pm 0.2$ | $12.2 \pm 0.7$ |
| YX06 | $6.6 \pm 0.3$ | $0.43 \pm 0.04$ | $5.4 \pm 0.5$ | $60 \pm 3$ | $1.5 \pm 0.4$ | $<$ | $9.9 \pm 0.6$ |
| YX07 | $12.7 \pm 0.4$ | $1.19 \pm 0.05$ | $16.2 \pm 0.7$ | $39 \pm 2$ | $1.2 \pm 0.3$ | $0.2 \pm 0.1$ | $20.7 \pm 0.8$ |
| YX08 | $10.6 \pm 0.4$ | $0.96 \pm 0.04$ | $14.1 \pm 0.6$ | $34 \pm 2$ | $1.3 \pm 0.3$ | $0.2 \pm 0.1$ | $17.2 \pm 0.7$ |
| YX09 | $22.2 \pm 0.8$ | $2.65 \pm 0.09$ | $31 \pm 1$ | $62 \pm 3$ | $2.2 \pm 0.5$ | $0.9 \pm 0.2$ | $45 \pm 2$ |
| YX10 | $23.0 \pm 0.8$ | $3.0 \pm 0.1$ | $31 \pm 1$ | $62 \pm 3$ | $2.7 \pm 0.5$ | $0.6 \pm 0.2$ | $47 \pm 2$ |
| YX11 | $16.3 \pm 0.6$ | $1.28 \pm 0.05$ | $18.7 \pm 0.8$ | $41 \pm 2$ | $1.9 \pm 0.3$ | $0.4 \pm 0.1$ | $24.7 \pm 0.9$ |
| YX12 | $13 \pm 0.4$ | $1.17 \pm 0.04$ | $16.03 \pm 0.7$ | $36 \pm 2$ | $1.5 \pm 0.3$ | $0.3 \pm 0.1$ | $21.5 \pm 0.8$ |
| YX13 | $48 \pm 2$ | $6.9 \pm 0.2$ | $71 \pm 3$ | $129 \pm 5$ | $3.8 \pm 0.6$ | $1.5 \pm 0.3$ | $86 \pm 3$ |
| YX14 | $51 \pm 2$ | $7.2 \pm 0.2$ | $72 \pm 3$ | $132 \pm 5$ | $4.3 \pm 0.5$ | $1.5 \pm 0.2$ | $90 \pm 3$ |
| YX15 | $47 \pm 1$ | $6.4 \pm 0.2$ | $65 \pm 2$ | $99 \pm 4$ | $1.4 \pm 0.3$ | $1.4 \pm 0.1$ | $38 \pm 1$ |
| YX16 | $45 \pm 1$ | $6.1 \pm 0.2$ | $62 \pm 2$ | $92 \pm 3$ | $1.2 \pm 0.3$ | $1.3 \pm 0.1$ | $34 \pm 1$ |
| YX17 | $348 \pm 11$ | $62 \pm 2$ | $446 \pm 14$ | $684 \pm 21$ | $11.4 \pm 0.7$ | $10.2 \pm 0.4$ | $318 \pm 10$ |
| YX18 | $319 \pm 10$ | $58 \pm 2$ | $411 \pm 13$ | $627 \pm 19$ | $10.6 \pm 0.7$ | $9.2 \pm 0.3$ | $295 \pm 9$ |
| YX19 | $30 \pm 1$ | $4.6 \pm 0.1$ | $38 \pm 1$ | $83 \pm 3$ | $3.7 \pm 0.4$ | $0.8 \pm 0.2$ | $39 \pm 1$ |
| YX20 | $25.6 \pm 0.8$ | $4.2 \pm 0.1$ | $34 \pm 1$ | $81 \pm 3$ | $3.4 \pm 0.3$ | $0.9 \pm 0.1$ | $34 \pm 1$ |
| YX21 | $17.8 \pm 0.7$ | $2.4 \pm 0.09$ | $24 \pm 1$ | $55 \pm 3$ | $4.1 \pm 0.6$ | $0.5 \pm 0.3$ | $27 \pm 1$ |
| YX22 | $15.9 \pm 0.6$ | $2.08 \pm 0.08$ | $21 \pm 1$ | $50 \pm 3$ | $3.8 \pm 0.5$ | $0.3 \pm 0.2$ | $25 \pm 1$ |
| YX23 | $4.2 \pm 0.2$ | $0.64 \pm 0.03$ | $6.9 \pm 0.4$ | $17 \pm 2$ | $2.8 \pm 0.3$ | $0.2 \pm 0.2$ | $11.9 \pm 0.5$ |
| YX24 | $3.1 \pm 0.2$ | $0.53 \pm 0.03$ | $4.8 \pm 0.3$ | $21 \pm 1$ | $2.7 \pm 0.3$ | $<$ | $9.1 \pm 0.4$ |
| YX25 | $23.1 \pm 0.8$ | $4.5 \pm 0.1$ | $49 \pm 2$ | $80 \pm 4$ | $3.9 \pm 0.5$ | $1.2 \pm 0.2$ | $43 \pm 2$ |
| YX26 | $21.0 \pm 0.7$ | $3.6 \pm 0.1$ | $45 \pm 2$ | $70 \pm 3$ | $2.9 \pm 0.4$ | $0.8 \pm 0.2$ | $38 \pm 1$ |
| YX27 | $7.5 \pm 0.3$ | $0.87 \pm 0.04$ | $11.3 \pm 0.6$ | $22 \pm 2$ | $2.0 \pm 0.3$ | $0.2 \pm 0.1$ | $11.5 \pm 0.5$ |
| YX28 | $5.4 \pm 0.2$ | $0.71 \pm 0.03$ | $8.7 \pm 0.5$ | $67 \pm 3$ | $1.8 \pm 0.3$ | $0.2 \pm 0.1$ | $8.9 \pm 0.4$ |
| YX29 | $52 \pm 2$ | $8.4 \pm 0.3$ | $79 \pm 3$ | $152 \pm 5$ | $3.5 \pm 0.3$ | $1.53 \pm 0.09$ | $69 \pm 2$ |
| YX30 | $46 \pm 1$ | $7.6 \pm 0.2$ | $71 \pm 2$ | $143 \pm 5$ | $3.2 \pm 0.2$ | $1.43 \pm 0.08$ | $62 \pm 2$ |
| YX31 | $29 \pm 1$ | $4.1 \pm 0.1$ | $41 \pm 2$ | $80 \pm 4$ | $3.8 \pm 0.6$ | $0.7 \pm 0.2$ | $44 \pm 2$ |
| YX32 | $25.8 \pm 0.9$ | $3.7 \pm 0.1$ | $37 \pm 1$ | $69 \pm 3$ | $3.1 \pm 0.4$ | $0.8 \pm 0.2$ | $37 \pm 1$ |
| YX33 | $9.6 \pm 0.4$ | $1.27 \pm 0.05$ | $12.3 \pm 0.7$ | $29 \pm 2$ | $2.1 \pm 0.4$ | $0.3 \pm 0.2$ | $15.6 \pm 0.7$ |
| YX34 | $8.3 \pm 0.3$ | $1.05 \pm 0.04$ | $10.3 \pm 0.6$ | $34 \pm 2$ | $2.3 \pm 0.4$ | $0.3 \pm 0.2$ | $13.1 \pm 0.6$ |





**Table A3.** Elemental air concentrations measured with ICP-AES, continued

| Element | Ti | Zn | Zr |
|---|---|---|---|
| Wavelenght (nm) | 334.187 | 213.86 | 339.2 |
| Analytical DL (ng) | 0.01 | 0.001 | 0.003 |
| Field DL (ng) | 0.2 | 0.1 | 0.01 |
| Sample name | $\mathrm{ng\,m^{-3}}$ | $\mathrm{ng\,m^{-3}}$ | $\mathrm{ng\,m^{-3}}$ |
| YX03 | $29 \pm 2$ | $9.1 \pm 0.4$ | $1.3 \pm 0.3$ |
| YX04 | $26 \pm 1$ | $11.3 \pm 0.5$ | $1.1 \pm 0.2$ |
| YX05 | $28 \pm 2$ | $11.9 \pm 0.6$ | < |
| YX06 | $20 \pm 2$ | $5.7 \pm 0.4$ | < |
| YX07 | $76 \pm 3$ | $16.7 \pm 0.6$ | $3.1 \pm 0.3$ |
| YX08 | $64 \pm 2$ | $61.7 \pm 2$ | $2.6 \pm 0.3$ |
| YX09 | $177 \pm 6$ | $11 \pm 0.6$ | $7.1 \pm 0.5$ |
| YX10 | $189 \pm 6$ | $9 \pm 0.5$ | $8 \pm 0.5$ |
| YX11 | $85 \pm 3$ | $25.3 \pm 0.9$ | $4.1 \pm 0.3$ |
| YX12 | $71 \pm 3$ | $25.5 \pm 0.9$ | $3.3 \pm 0.3$ |
| YX13 | $430 \pm 14$ | $29 \pm 1$ | $17.3 \pm 0.7$ |
| YX14 | $438 \pm 14$ | $17.7 \pm 0.7$ | $17.5 \pm 0.7$ |
| YX15 | $452 \pm 14$ | $19.4 \pm 0.7$ | $17.8 \pm 0.6$ |
| YX16 | $431 \pm 13$ | $12.1 \pm 0.5$ | $16.7 \pm 0.6$ |
| YX17 | $3145 \pm 95$ | $75 \pm 2$ | $124 \pm 4$ |
| YX18 | $2871 \pm 87$ | $71 \pm 2$ | $111 \pm 3$ |
| YX19 | $275 \pm 9$ | $14.4 \pm 0.6$ | $10.8 \pm 0.5$ |
| YX20 | $228 \pm 7$ | $15.1 \pm 0.6$ | $9.5 \pm 0.4$ |
| YX21 | $157 \pm 6$ | $10.4 \pm 0.6$ | $5.5 \pm 0.5$ |
| YX22 | $129 \pm 5$ | $5.3 \pm 0.4$ | $5 \pm 0.5$ |
| YX23 | $32 \pm 2$ | $7.5 \pm 0.4$ | $1.5 \pm 0.3$ |
| YX24 | $19 \pm 1$ | $6.8 \pm 0.3$ | $1.1 \pm 0.2$ |
| YX25 | $336 \pm 11$ | $18.4 \pm 0.8$ | $15.9 \pm 0.6$ |
| YX26 | $266 \pm 9$ | $9.5 \pm 0.5$ | $12.2 \pm 0.5$ |
| YX27 | $68 \pm 3$ | $8.3 \pm 0.4$ | $2.9 \pm 0.3$ |
| YX28 | $46 \pm 2$ | $7.4 \pm 0.3$ | $2.1 \pm 0.2$ |
| YX29 | $488 \pm 15$ | $37 \pm 1$ | $21.1 \pm 0.7$ |
| YX30 | $440 \pm 13$ | $34 \pm 1$ | $19.1 \pm 0.6$ |
| YX31 | $265 \pm 9$ | $27 \pm 1$ | $11 \pm 0.6$ |
| YX32 | $236 \pm 8$ | $25.2 \pm 0.9$ | $9.5 \pm 0.5$ |
| YX33 | $73 \pm 3$ | $9.5 \pm 0.5$ | $3.2 \pm 0.4$ |
| YX34 | $58 \pm 2$ | $10 \pm 0.5$ | $2.1 \pm 0.3$ |



**Table A4.** Elemental air concentrations measured with ICP-MS.

| Element | Be | Cd | Co | Cr | Cu | Mo | Ni |
|---|---|---|---|---|---|---|---|
| Isotope | 9 | 111 | 59 | 52 | 63 | 95 | 60 |
| Analytical DL (ng) | 0.02 | 0.04 | 0.04 | 0.4 | 0.1 | 0.3 | 3 |
| Field DL (ng) | 0.1 | 0.1 | 2 | 643 | 15 | 3 | 141 |
| Sample name | $\mathrm{ng\,m^{-3}}$ | $\mathrm{ng\,m^{-3}}$ | $\mathrm{ng\,m^{-3}}$ | $\mathrm{ng\,m^{-3}}$ | $\mathrm{ng\,m^{-3}}$ | $\mathrm{ng\,m^{-3}}$ | $\mathrm{ng\,m^{-3}}$ |
| YX03 | 0.018 ± 0.003 | 0.06 ± 0.01 | 0.19 ± 0.02 | < | 1.28 ± 0.1 | < | < |
| YX04 | 0.012 ± 0.002 | 0.06 ± 0.01 | 0.13 ± 0.02 | < | 1.31 ± 0.1 | < | < |
| YX05 | 0.03 ± 0.01 | 0.09 ± 0.01 | 0.27 ± 0.04 | < | 1.6 ± 0.2 | < | < |
| YX06 | 0.015 ± 0.004 | 0.08 ± 0.01 | 0.15 ± 0.03 | < | 1.3 ± 0.1 | < | < |
| YX07 | 0.05 ± 0.01 | 0.15 ± 0.01 | 0.36 ± 0.03 | < | 3.4 ± 0.2 | < | < |
| YX08 | 0.03 ± 0.01 | 0.14 ± 0.01 | 0.31 ± 0.03 | < | 2.9 ± 0.2 | < | 1 ± 2 |
| YX09 | 0.11 ± 0.02 | 0.07 ± 0.01 | 0.75 ± 0.07 | < | 1.9 ± 0.2 | < | < |
| YX10 | 0.11 ± 0.01 | 0.07 ± 0.01 | 0.83 ± 0.07 | < | 2.5 ± 0.2 | < | 3 ± 3 |
| YX11 | 0.04 ± 0.01 | 0.22 ± 0.02 | 0.44 ± 0.04 | < | 4.3 ± 0.2 | 0.19 ± 0.04 | 3 ± 2 |
| YX12 | 0.04 ± 0.01 | 0.2 ± 0.02 | 0.35 ± 0.03 | < | 3.1 ± 0.2 | 0.15 ± 0.04 | 2 ± 2 |
| YX13 | 0.14 ± 0.02 | 0.05 ± 0.01 | 1.01 ± 0.08 | < | 2 ± 0.1 | 0.15 ± 0.04 | 2 ± 2 |
| YX14 | 0.28 ± 0.04 | 0.08 ± 0.01 | 2 ± 0.2 | 10 ± 3 | 4.1 ± 0.3 | 0.35 ± 0.07 | 6 ± 4 |
| YX15 | 0.25 ± 0.03 | 0.04 ± 0.005 | 1.7 ± 0.1 | < | 3.1 ± 0.2 | 0.19 ± 0.04 | 3 ± 2 |
| YX16 | 0.24 ± 0.03 | 0.04 ± 0.005 | 1.6 ± 0.1 | 7 ± 2 | 2.9 ± 0.2 | 0.16 ± 0.04 | 3 ± 2 |
| YX17 | 1.7 ± 0.2 | 0.24 ± 0.02 | 11.4 ± 0.7 | 48 ± 4 | 17.6 ± 0.8 | 1.5 ± 0.2 | 26 ± 6 |
| YX18 | 1.7 ± 0.2 | 0.22 ± 0.02 | 11.2 ± 0.7 | 42 ± 4 | 16.4 ± 0.7 | 1.2 ± 0.1 | 24 ± 6 |
| YX19 | 0.17 ± 0.02 | 0.06 ± 0.01 | 1.11 ± 0.08 | < | 2.3 ± 0.2 | 0.17 ± 0.05 | 4 ± 2 |
| YX20 | 0.14 ± 0.02 | 0.05 ± 0.01 | 1.06 ± 0.08 | 6 ± 2 | 2 ± 0.1 | 0.18 ± 0.04 | 4 ± 2 |
| YX21 | 0.08 ± 0.01 | 0.09 ± 0.01 | 0.61 ± 0.06 | < | 1.5 ± 0.2 | < | 4 ± 4 |
| YX22 | 0.07 ± 0.01 | 0.09 ± 0.01 | 1.2 ± 0.1 | < | 1.4 ± 0.1 | 0.35 ± 0.07 | 5 ± 4 |
| YX23 | 0.026 ± 0.004 | 0.04 ± 0.01 | 0.2 ± 0.02 | < | 1.1 ± 0.1 | < | 4 ± 2 |
| YX24 | 0.014 ± 0.003 | 0.04 ± 0.005 | 0.15 ± 0.02 | < | 1.1 ± 0.1 | 0.12 ± 0.03 | 3 ± 2 |
| YX25 | 0.15 ± 0.02 | 0.07 ± 0.01 | 1.1 ± 0.09 | < | 2.5 ± 0.2 | 0.24 ± 0.06 | 2 ± 3 |
| YX26 | 0.16 ± 0.02 | 0.06 ± 0.01 | 0.99 ± 0.08 | < | 2.2 ± 0.2 | 0.22 ± 0.05 | 4 ± 3 |
| YX27 | 0.03 ± 0.01 | 0.05 ± 0.01 | 0.28 ± 0.03 | < | 1.4 ± 0.1 | 0.16 ± 0.04 | 2 ± 2 |
| YX28 | 0.028 ± 0.004 | 0.05 ± 0.01 | 0.22 ± 0.02 | < | 1.1 ± 0.1 | < | 2 ± 2 |
| YX29 | 0.33 ± 0.04 | 0.13 ± 0.01 | 2 ± 0.1 | 10 ± 1 | 4.5 ± 0.2 | 0.35 ± 0.04 | 6 ± 2 |
| YX30 | 0.29 ± 0.04 | 0.12 ± 0.01 | 1.8 ± 0.1 | 9 ± 1 | 4.8 ± 0.2 | 0.33 ± 0.04 | 6 ± 2 |
| YX31 | 0.17 ± 0.03 | 0.14 ± 0.02 | 1.1 ± 0.1 | < | 6.7 ± 0.4 | < | 4 ± 4 |
| YX32 | 0.14 ± 0.02 | 0.12 ± 0.01 | 1.04 ± 0.07 | < | 2.7 ± 0.2 | < | 3 ± 3 |
| YX33 | 0.04 ± 0.01 | 0.06 ± 0.01 | 0.36 ± 0.04 | < | 1.8 ± 0.1 | < | 2 ± 2 |
| YX34 | 0.03 ± 0.01 | 0.06 ± 0.01 | 0.36 ± 0.04 | < | 1.6 ± 0.1 | < | 2 ± 2 |


**Table A5.** Elemental air concentrations measured with ICP-MS, continued.

| Element | Pb | Rb | Sb | Se | Tl | U | V |
|---|---|---|---|---|---|---|---|
| Isotope | 208 | 85 | 121 | 77 | 205 | 238 | 51 |
| Analytical DL (ng) | 0.01 | 0.1 | 0.02 | 0.5 | 0.002 | 0.01 | 0.4 |
| Field DL (ng) | 2 | 4 | 0.6 | — | 0.1 | 0.04 | 2 |
| Sample name | $\mathrm{ng\,m^{-3}}$ | $\mathrm{ng\,m^{-3}}$ | $\mathrm{ng\,m^{-3}}$ | $\mathrm{ng\,m^{-3}}$ | $\mathrm{ng\,m^{-3}}$ | $\mathrm{ng\,m^{-3}}$ | $\mathrm{ng\,m^{-3}}$ |
| YX03 | $4 \pm 1.1$ | $1.04 \pm 0.05$ | $0.6 \pm 0.04$ | $0.4 \pm 0.1$ | $0.017 \pm 0.003$ | $0.04 \pm 0.01$ | $2.8 \pm 0.1$ |
| YX04 | $3.3 \pm 0.9$ | $0.98 \pm 0.05$ | $0.55 \pm 0.04$ | $0.29 \pm 0.09$ | $0.014 \pm 0.003$ | $0.03 \pm 0.01$ | $2.9 \pm 0.1$ |
| YX05 | $3.4 \pm 0.9$ | $1.11 \pm 0.07$ | $0.51 \pm 0.04$ | $0.18 \pm 0.09$ | $0.011 \pm 0.003$ | $0.03 \pm 0.01$ | $2.2 \pm 0.1$ |
| YX06 | $3.3 \pm 0.9$ | $1.05 \pm 0.06$ | $0.49 \pm 0.04$ | $0.17 \pm 0.08$ | $0.012 \pm 0.003$ | $0.04 \pm 0.01$ | $2.1 \pm 0.1$ |
| YX07 | $4.9 \pm 1.3$ | $2 \pm 0.09$ | $1.2 \pm 0.08$ | $0.14 \pm 0.06$ | $0.023 \pm 0.004$ | $0.07 \pm 0.01$ | $3.3 \pm 0.2$ |
| YX08 | $4.4 \pm 1.2$ | $1.63 \pm 0.07$ | $1.12 \pm 0.07$ | $0.14 \pm 0.06$ | $0.02 \pm 0.004$ | $0.05 \pm 0.01$ | $2.7 \pm 0.1$ |
| YX09 | $3.4 \pm 1$ | $4.0 \pm 0.2$ | $0.43 \pm 0.03$ | $<$ | $0.03 \pm 0.01$ | $0.14 \pm 0.03$ | $5.6 \pm 0.3$ |
| YX10 | $3.5 \pm 0.9$ | $4.3 \pm 0.2$ | $0.38 \pm 0.03$ | $0.09 \pm 0.07$ | $0.03 \pm 0.01$ | $0.14 \pm 0.03$ | $5.9 \pm 0.3$ |
| YX11 | $5.9 \pm 1.5$ | $2.3 \pm 0.1$ | $1.11 \pm 0.07$ | $0.3 \pm 0.09$ | $0.027 \pm 0.005$ | $0.08 \pm 0.02$ | $3.6 \pm 0.2$ |
| YX12 | $5.2 \pm 1.4$ | $2.01 \pm 0.09$ | $1.03 \pm 0.07$ | $0.22 \pm 0.07$ | $0.024 \pm 0.004$ | $0.07 \pm 0.01$ | $3.1 \pm 0.1$ |
| YX13 | $2 \pm 0.6$ | $5.6 \pm 0.2$ | $0.28 \pm 0.02$ | $0.12 \pm 0.06$ | $0.03 \pm 0.01$ | $0.13 \pm 0.03$ | $6.4 \pm 0.3$ |
| YX14 | $4 \pm 1.1$ | $11.7 \pm 0.5$ | $0.53 \pm 0.04$ | $0.3 \pm 0.1$ | $0.07 \pm 0.01$ | $0.28 \pm 0.06$ | $13.6 \pm 0.6$ |
| YX15 | $3.1 \pm 0.8$ | $8.7 \pm 0.3$ | $0.4 \pm 0.03$ | $0.18 \pm 0.06$ | $0.05 \pm 0.01$ | $0.28 \pm 0.06$ | $11.2 \pm 0.5$ |
| YX16 | $2.6 \pm 0.7$ | $8 \pm 0.3$ | $0.36 \pm 0.03$ | $0.15 \pm 0.06$ | $0.05 \pm 0.01$ | $0.24 \pm 0.05$ | $10.3 \pm 0.5$ |
| YX17 | $15.3 \pm 4$ | $63 \pm 2$ | $0.45 \pm 0.03$ | $0.4 \pm 0.1$ | $0.33 \pm 0.05$ | $1.9 \pm 0.4$ | $80 \pm 4$ |
| YX18 | $13 \pm 3$ | $60 \pm 2$ | $0.4 \pm 0.03$ | $0.4 \pm 0.1$ | $0.31 \pm 0.05$ | $1.6 \pm 0.3$ | $77 \pm 3$ |
| YX19 | $3.2 \pm 0.9$ | $5.7 \pm 0.2$ | $0.22 \pm 0.02$ | $0.5 \pm 0.2$ | $0.05 \pm 0.01$ | $0.16 \pm 0.04$ | $11.9 \pm 0.5$ |
| YX20 | $2.6 \pm 0.7$ | $4.9 \pm 0.2$ | $0.21 \pm 0.02$ | $0.6 \pm 0.2$ | $0.05 \pm 0.01$ | $0.14 \pm 0.03$ | $10.7 \pm 0.5$ |
| YX21 | $2.3 \pm 0.7$ | $3.3 \pm 0.1$ | $<$ | $0.9 \pm 0.2$ | $0.03 \pm 0.01$ | $0.11 \pm 0.02$ | $11.2 \pm 0.5$ |
| YX22 | $2 \pm 0.6$ | $3 \pm 0.1$ | $<$ | $0.5 \pm 0.1$ | $0.03 \pm 0.01$ | $0.1 \pm 0.02$ | $9.9 \pm 0.5$ |
| YX23 | $1.5 \pm 0.4$ | $1.02 \pm 0.05$ | $0.23 \pm 0.02$ | $0.6 \pm 0.2$ | $0.01 \pm 0.002$ | $0.02 \pm 0.01$ | $9.7 \pm 0.4$ |
| YX24 | $1.5 \pm 0.4$ | $0.83 \pm 0.05$ | $0.21 \pm 0.02$ | $0.8 \pm 0.2$ | $0.011 \pm 0.002$ | $0.017 \pm 0.005$ | $9.2 \pm 0.5$ |
| YX25 | $4.2 \pm 1.1$ | $5.9 \pm 0.2$ | $<$ | $0.72 \pm 0.2$ | $0.05 \pm 0.01$ | $0.18 \pm 0.04$ | $11.8 \pm 0.5$ |
| YX26 | $3.7 \pm 1$ | $5.6 \pm 0.2$ | $<$ | $0.5 \pm 0.2$ | $0.05 \pm 0.01$ | $0.14 \pm 0.03$ | $11.6 \pm 0.5$ |
| YX27 | $2.7 \pm 0.7$ | $1.37 \pm 0.06$ | $0.34 \pm 0.02$ | $0.5 \pm 0.1$ | $0.017 \pm 0.003$ | $0.05 \pm 0.01$ | $7.4 \pm 0.3$ |
| YX28 | $2.6 \pm 0.7$ | $1.09 \pm 0.05$ | $0.33 \pm 0.02$ | $0.7 \pm 0.2$ | $0.017 \pm 0.003$ | $0.03 \pm 0.01$ | $6.7 \pm 0.3$ |
| YX29 | $6 \pm 1$ | $10.7 \pm 0.4$ | $0.56 \pm 0.04$ | $0.6 \pm 0.2$ | $0.07 \pm 0.01$ | $0.25 \pm 0.06$ | $18.1 \pm 0.8$ |
| YX30 | $6 \pm 1$ | $10.4 \pm 0.4$ | $0.54 \pm 0.03$ | $0.5 \pm 0.1$ | $0.06 \pm 0.01$ | $0.24 \pm 0.05$ | $16.8 \pm 0.8$ |
| YX31 | $5 \pm 2$ | $5.8 \pm 0.3$ | $<$ | $0.6 \pm 0.2$ | $0.04 \pm 0.01$ | $0.15 \pm 0.04$ | $10.6 \pm 0.6$ |
| YX32 | $5 \pm 1$ | $5.4 \pm 0.2$ | $0.33 \pm 0.02$ | $0.5 \pm 0.1$ | $0.04 \pm 0.01$ | $0.14 \pm 0.03$ | $9.3 \pm 0.4$ |
| YX33 | $3.3 \pm 0.9$ | $1.80 \pm 0.08$ | $<$ | $0.8 \pm 0.2$ | $0.024 \pm 0.004$ | $0.05 \pm 0.01$ | $6.2 \pm 0.3$ |
| YX34 | $2.8 \pm 0.8$ | $1.46 \pm 0.07$ | $0.26 \pm 0.02$ | $0.8 \pm 0.2$ | $0.017 \pm 0.003$ | $0.04 \pm 0.01$ | $5.3 \pm 0.3$ |





**Table A6.** REE air concentrations measured with ICP-MS.

| Element | La | Ce | Pr | Nd | Sm | Eu | Gd |
|---|---|---|---|---|---|---|---|
| Isotope | 139 | 140 | 141 | 146 | 147 | 153 | 157 |
| Analytical DL (ng) | 0.01 | 0.004 | 0.001 | 0.01 | 0.01 | 0.01 | 0.004 |
| Field DL (ng) | 0.4 | 0.5 | 0.1 | 0.2 | 0.05 | 0.03 | 0.1 |
| Sample name | $ng\,m^{-3}$ | $ng\,m^{-3}$ | $ng\,m^{-3}$ | $ng\,m^{-3}$ | $ng\,m^{-3}$ | $ng\,m^{-3}$ | $ng\,m^{-3}$ |
| YX03 | $0.37 \pm 0.07$ | $0.8 \pm 0.2$ | $0.09 \pm 0.02$ | $0.3 \pm 0.1$ | $0.07 \pm 0.02$ | $0.013 \pm 0.004$ | $0.05 \pm 0.01$ |
| YX04 | $0.32 \pm 0.06$ | $0.6 \pm 0.1$ | $0.07 \pm 0.01$ | $0.3 \pm 0.1$ | $0.06 \pm 0.02$ | $0.011 \pm 0.003$ | $0.05 \pm 0.01$ |
| YX05 | $0.41 \pm 0.08$ | $0.8 \pm 0.2$ | $0.1 \pm 0.02$ | $0.4 \pm 0.1$ | $0.09 \pm 0.02$ | $0.016 \pm 0.005$ | $0.04 \pm 0.01$ |
| YX06 | $0.36 \pm 0.07$ | $0.7 \pm 0.2$ | $0.08 \pm 0.02$ | $0.3 \pm 0.1$ | $0.07 \pm 0.02$ | $0.012 \pm 0.004$ | $0.04 \pm 0.01$ |
| YX07 | $0.9 \pm 0.2$ | $1.7 \pm 0.4$ | $0.19 \pm 0.03$ | $0.8 \pm 0.3$ | $0.13 \pm 0.03$ | $0.028 \pm 0.009$ | $0.11 \pm 0.03$ |
| YX08 | $0.7 \pm 0.1$ | $1.4 \pm 0.3$ | $0.16 \pm 0.03$ | $0.6 \pm 0.2$ | $0.12 \pm 0.03$ | $0.024 \pm 0.007$ | $0.1 \pm 0.03$ |
| YX09 | $2.0 \pm 0.3$ | $3.8 \pm 0.8$ | $0.45 \pm 0.08$ | $1.9 \pm 0.7$ | $0.33 \pm 0.08$ | $0.08 \pm 0.02$ | $0.27 \pm 0.07$ |
| YX10 | $2.0 \pm 0.3$ | $4.0 \pm 0.9$ | $0.47 \pm 0.08$ | $1.8 \pm 0.7$ | $0.36 \pm 0.09$ | $0.07 \pm 0.02$ | $0.28 \pm 0.07$ |
| YX11 | $0.9 \pm 0.2$ | $2 \pm 0.4$ | $0.22 \pm 0.04$ | $0.9 \pm 0.3$ | $0.18 \pm 0.04$ | $0.04 \pm 0.01$ | $0.13 \pm 0.03$ |
| YX12 | $0.8 \pm 0.1$ | $1.6 \pm 0.3$ | $0.19 \pm 0.03$ | $0.7 \pm 0.3$ | $0.15 \pm 0.04$ | $0.031 \pm 0.009$ | $0.11 \pm 0.03$ |
| YX13 | $2.2 \pm 0.4$ | $4.4 \pm 0.9$ | $0.53 \pm 0.09$ | $1.9 \pm 0.7$ | $0.4 \pm 0.1$ | $0.07 \pm 0.02$ | $0.29 \pm 0.07$ |
| YX14 | $4 \pm 0.7$ | $7 \pm 2$ | $0.9 \pm 0.2$ | $4 \pm 1$ | $0.7 \pm 0.2$ | $0.15 \pm 0.05$ | $0.6 \pm 0.1$ |
| YX15 | $4.7 \pm 0.8$ | $10 \pm 2$ | $1.1 \pm 0.2$ | $4 \pm 2$ | $0.8 \pm 0.2$ | $0.16 \pm 0.05$ | $0.6 \pm 0.1$ |
| YX16 | $4.2 \pm 0.7$ | $9 \pm 2$ | $0.9 \pm 0.2$ | $4 \pm 1$ | $0.7 \pm 0.2$ | $0.14 \pm 0.04$ | $0.6 \pm 0.1$ |
| YX17 | $31 \pm 5$ | $60 \pm 10$ | $7 \pm 1$ | $24 \pm 9$ | $5 \pm 1$ | $1 \pm 0.3$ | $4.1 \pm 1$ |
| YX18 | $28 \pm 5$ | $60 \pm 10$ | $7 \pm 1$ | $25 \pm 9$ | $5 \pm 1$ | $0.9 \pm 0.3$ | $3.7 \pm 0.9$ |
| YX19 | $2.6 \pm 0.4$ | $5 \pm 1$ | $0.6 \pm 0.1$ | $2 \pm 0.7$ | $0.4 \pm 0.1$ | $0.08 \pm 0.02$ | $0.33 \pm 0.08$ |
| YX20 | $2.5 \pm 0.4$ | $5 \pm 1$ | $0.56 \pm 0.09$ | $2.2 \pm 0.8$ | $0.4 \pm 0.1$ | $0.08 \pm 0.02$ | $0.33 \pm 0.08$ |
| YX21 | $1.5 \pm 0.3$ | $3.1 \pm 0.7$ | $0.33 \pm 0.06$ | $1.4 \pm 0.5$ | $0.28 \pm 0.07$ | $0.05 \pm 0.02$ | $0.19 \pm 0.05$ |
| YX22 | $1.5 \pm 0.3$ | $3 \pm 0.6$ | $0.34 \pm 0.06$ | $1.4 \pm 0.5$ | $0.26 \pm 0.06$ | $0.06 \pm 0.02$ | $0.19 \pm 0.05$ |
| YX23 | $0.36 \pm 0.07$ | $0.7 \pm 0.2$ | $0.08 \pm 0.01$ | $0.3 \pm 0.1$ | $0.05 \pm 0.01$ | $0.013 \pm 0.004$ | $0.06 \pm 0.02$ |
| YX24 | $0.27 \pm 0.05$ | $0.5 \pm 0.1$ | $0.06 \pm 0.01$ | $0.23 \pm 0.09$ | $0.04 \pm 0.01$ | $0.008 \pm 0.003$ | $0.034 \pm 0.009$ |
| YX25 | $3.2 \pm 0.5$ | $6 \pm 1$ | $0.7 \pm 0.1$ | $3 \pm 1$ | $0.6 \pm 0.1$ | $0.14 \pm 0.04$ | $0.5 \pm 0.1$ |
| YX26 | $2.7 \pm 0.5$ | $6 \pm 1$ | $0.6 \pm 0.1$ | $2.5 \pm 1$ | $0.5 \pm 0.1$ | $0.1 \pm 0.03$ | $0.4 \pm 0.1$ |
| YX27 | $0.7 \pm 0.1$ | $1.3 \pm 0.3$ | $0.15 \pm 0.03$ | $0.6 \pm 0.2$ | $0.12 \pm 0.03$ | $0.026 \pm 0.008$ | $0.08 \pm 0.02$ |
| YX28 | $0.52 \pm 0.09$ | $1 \pm 0.2$ | $0.12 \pm 0.02$ | $0.5 \pm 0.2$ | $0.09 \pm 0.02$ | $0.02 \pm 0.006$ | $0.07 \pm 0.02$ |
| YX29 | $4.6 \pm 0.8$ | $9 \pm 2$ | $1.1 \pm 0.2$ | $4 \pm 1$ | $0.8 \pm 0.2$ | $0.15 \pm 0.04$ | $0.6 \pm 0.2$ |
| YX30 | $4.3 \pm 0.7$ | $9 \pm 2$ | $1.0 \pm 0.2$ | $4 \pm 1$ | $0.8 \pm 0.2$ | $0.15 \pm 0.04$ | $0.6 \pm 0.1$ |
| YX31 | $2.7 \pm 0.5$ | $5 \pm 1$ | $0.6 \pm 0.1$ | $2.1 \pm 0.8$ | $0.5 \pm 0.1$ | $0.09 \pm 0.03$ | $0.34 \pm 0.09$ |
| YX32 | $2.2 \pm 0.4$ | $4.5 \pm 1$ | $0.51 \pm 0.09$ | $1.9 \pm 0.7$ | $0.37 \pm 0.09$ | $0.08 \pm 0.02$ | $0.26 \pm 0.07$ |
| YX33 | $0.8 \pm 0.1$ | $1.7 \pm 0.4$ | $0.19 \pm 0.03$ | $0.8 \pm 0.3$ | $0.14 \pm 0.04$ | $0.028 \pm 0.009$ | $0.12 \pm 0.03$ |
| YX34 | $0.7 \pm 0.1$ | $1.3 \pm 0.3$ | $0.14 \pm 0.02$ | $0.6 \pm 0.2$ | $0.1 \pm 0.03$ | $0.021 \pm 0.006$ | $0.08 \pm 0.02$ |



**Table A7.** REE air concentrations measured with ICP-MS, continued.

| Element | Tb | Dy | Ho | Er | Tm | Yb | Lu |
|---|---|---|---|---|---|---|---|
| Isotope | 159 | 163 | 165 | 166 | 169 | 172 | 175 |
| Analytical DL (ng) | 0.003 | 0.004 | 0.001 | 0.001 | 0.01 | 0.002 | 0.001 |
| Field DL (ng) | 0.02 | 0.05 | 0.01 | 0.02 | — | 0.04 | 0.01 |
| Sample name | $\mathrm{ng\,m^{-3}}$ | $\mathrm{ng\,m^{-3}}$ | $\mathrm{ng\,m^{-3}}$ | $\mathrm{ng\,m^{-3}}$ | $\mathrm{ng\,m^{-3}}$ | $\mathrm{ng\,m^{-3}}$ | $\mathrm{ng\,m^{-3}}$ |
| YX03 | $0.007 \pm 0.002$ | $0.05 \pm 0.01$ | $0.009 \pm 0.002$ | $0.027 \pm 0.007$ | $0.004 \pm 0.002$ | $0.019 \pm 0.005$ | $0.003 \pm 0.0008$ |
| YX04 | $0.007 \pm 0.002$ | $0.04 \pm 0.01$ | $0.008 \pm 0.002$ | $0.019 \pm 0.005$ | $0.003 \pm 0.002$ | $0.018 \pm 0.004$ | $0.0019 \pm 0.0005$ |
| YX05 | $0.007 \pm 0.002$ | $0.05 \pm 0.02$ | $0.008 \pm 0.002$ | $0.026 \pm 0.007$ | $0.006 \pm 0.003$ | $0.025 \pm 0.006$ | $0.004 \pm 0.001$ |
| YX06 | $0.007 \pm 0.002$ | $0.04 \pm 0.01$ | $0.007 \pm 0.002$ | $0.024 \pm 0.007$ | $0.005 \pm 0.003$ | $0.023 \pm 0.006$ | $0.005 \pm 0.001$ |
| YX07 | $0.019 \pm 0.004$ | $0.11 \pm 0.03$ | $0.019 \pm 0.004$ | $0.05 \pm 0.01$ | $0.007 \pm 0.002$ | $0.05 \pm 0.01$ | $0.007 \pm 0.002$ |
| YX08 | $0.013 \pm 0.003$ | $0.08 \pm 0.02$ | $0.016 \pm 0.003$ | $0.05 \pm 0.01$ | $0.007 \pm 0.002$ | $0.034 \pm 0.008$ | $0.006 \pm 0.001$ |
| YX09 | $0.06 \pm 0.01$ | $0.23 \pm 0.07$ | $0.042 \pm 0.009$ | $0.12 \pm 0.03$ | $0.019 \pm 0.005$ | $0.10 \pm 0.02$ | $0.014 \pm 0.003$ |
| YX10 | $0.042 \pm 0.009$ | $0.24 \pm 0.07$ | $0.045 \pm 0.009$ | $0.13 \pm 0.04$ | $0.02 \pm 0.006$ | $0.12 \pm 0.03$ | $0.017 \pm 0.004$ |
| YX11 | $0.02 \pm 0.004$ | $0.14 \pm 0.04$ | $0.023 \pm 0.005$ | $0.06 \pm 0.02$ | $0.01 \pm 0.003$ | $0.05 \pm 0.01$ | $0.008 \pm 0.002$ |
| YX12 | $0.015 \pm 0.003$ | $0.1 \pm 0.03$ | $0.018 \pm 0.004$ | $0.06 \pm 0.02$ | $0.006 \pm 0.002$ | $0.05 \pm 0.01$ | $0.007 \pm 0.002$ |
| YX13 | $0.045 \pm 0.009$ | $0.24 \pm 0.07$ | $0.049 \pm 0.01$ | $0.13 \pm 0.04$ | $0.018 \pm 0.005$ | $0.13 \pm 0.03$ | $0.017 \pm 0.004$ |
| YX14 | $0.08 \pm 0.02$ | $0.5 \pm 0.1$ | $0.09 \pm 0.02$ | $0.26 \pm 0.07$ | $0.032 \pm 0.009$ | $0.2 \pm 0.05$ | $0.028 \pm 0.007$ |
| YX15 | $0.09 \pm 0.02$ | $0.5 \pm 0.2$ | $0.09 \pm 0.02$ | $0.28 \pm 0.07$ | $0.038 \pm 0.01$ | $0.25 \pm 0.06$ | $0.034 \pm 0.008$ |
| YX16 | $0.07 \pm 0.01$ | $0.5 \pm 0.1$ | $0.09 \pm 0.02$ | $0.25 \pm 0.07$ | $0.034 \pm 0.009$ | $0.19 \pm 0.04$ | $0.029 \pm 0.007$ |
| YX17 | $0.6 \pm 0.1$ | $3 \pm 1$ | $0.7 \pm 0.1$ | $1.7 \pm 0.5$ | $0.27 \pm 0.07$ | $1.5 \pm 0.3$ | $0.23 \pm 0.05$ |
| YX18 | $0.5 \pm 0.1$ | $3 \pm 1$ | $0.6 \pm 0.1$ | $1.7 \pm 0.5$ | $0.22 \pm 0.06$ | $1.5 \pm 0.3$ | $0.22 \pm 0.05$ |
| YX19 | $0.05 \pm 0.01$ | $0.3 \pm 0.09$ | $0.05 \pm 0.01$ | $0.15 \pm 0.04$ | $0.022 \pm 0.006$ | $0.13 \pm 0.03$ | $0.019 \pm 0.005$ |
| YX20 | $0.045 \pm 0.009$ | $0.29 \pm 0.09$ | $0.05 \pm 0.01$ | $0.14 \pm 0.04$ | $0.018 \pm 0.005$ | $0.13 \pm 0.03$ | $0.019 \pm 0.004$ |
| YX21 | $0.029 \pm 0.006$ | $0.18 \pm 0.06$ | $0.033 \pm 0.007$ | $0.1 \pm 0.03$ | $0.012 \pm 0.004$ | $0.09 \pm 0.02$ | $0.011 \pm 0.003$ |
| YX22 | $0.032 \pm 0.007$ | $0.18 \pm 0.06$ | $0.034 \pm 0.007$ | $0.09 \pm 0.02$ | $0.013 \pm 0.004$ | $0.08 \pm 0.02$ | $0.013 \pm 0.003$ |
| YX23 | $0.004 \pm 0.001$ | $0.05 \pm 0.01$ | $0.009 \pm 0.002$ | $0.024 \pm 0.007$ | $0.004 \pm 0.002$ | $0.021 \pm 0.005$ | $0.0022 \pm 0.0006$ |
| YX24 | $0.005 \pm 0.001$ | $0.03 \pm 0.01$ | $0.006 \pm 0.001$ | $0.011 \pm 0.003$ | $0.002 \pm 0.001$ | $0.017 \pm 0.004$ | $0.0024 \pm 0.0007$ |
| YX25 | $0.06 \pm 0.01$ | $0.4 \pm 0.1$ | $0.08 \pm 0.01$ | $0.21 \pm 0.06$ | $0.03 \pm 0.008$ | $0.19 \pm 0.04$ | $0.028 \pm 0.007$ |
| YX26 | $0.06 \pm 0.01$ | $0.32 \pm 0.1$ | $0.06 \pm 0.01$ | $0.17 \pm 0.04$ | $0.026 \pm 0.007$ | $0.15 \pm 0.03$ | $0.022 \pm 0.005$ |
| YX27 | $0.012 \pm 0.003$ | $0.09 \pm 0.03$ | $0.015 \pm 0.003$ | $0.05 \pm 0.01$ | $0.006 \pm 0.002$ | $0.03 \pm 0.007$ | $0.005 \pm 0.001$ |
| YX28 | $0.009 \pm 0.002$ | $0.06 \pm 0.02$ | $0.012 \pm 0.002$ | $0.035 \pm 0.01$ | $0.005 \pm 0.002$ | $0.028 \pm 0.007$ | $0.004 \pm 0.001$ |
| YX29 | $0.09 \pm 0.02$ | $0.5 \pm 0.2$ | $0.1 \pm 0.02$ | $0.26 \pm 0.07$ | $0.035 \pm 0.009$ | $0.24 \pm 0.05$ | $0.033 \pm 0.008$ |
| YX30 | $0.08 \pm 0.02$ | $0.5 \pm 0.2$ | $0.09 \pm 0.02$ | $0.27 \pm 0.07$ | $0.035 \pm 0.009$ | $0.22 \pm 0.05$ | $0.032 \pm 0.008$ |
| YX31 | $0.05 \pm 0.01$ | $0.3 \pm 0.1$ | $0.05 \pm 0.01$ | $0.17 \pm 0.04$ | $0.022 \pm 0.006$ | $0.14 \pm 0.03$ | $0.022 \pm 0.005$ |
| YX32 | $0.039 \pm 0.008$ | $0.28 \pm 0.09$ | $0.048 \pm 0.01$ | $0.12 \pm 0.03$ | $0.019 \pm 0.005$ | $0.11 \pm 0.02$ | $0.014 \pm 0.003$ |
| YX33 | $0.014 \pm 0.003$ | $0.08 \pm 0.03$ | $0.018 \pm 0.004$ | $0.05 \pm 0.01$ | $0.007 \pm 0.003$ | $0.05 \pm 0.01$ | $0.007 \pm 0.002$ |
| YX34 | $0.013 \pm 0.003$ | $0.07 \pm 0.02$ | $0.015 \pm 0.003$ | $0.04 \pm 0.01$ | $0.006 \pm 0.002$ | $0.035 \pm 0.008$ | $0.006 \pm 0.001$ |





| Analysed by ICP-AES | Measured µg/ g or mg/g | Recovery rate | Analysed by ICP-MS | Measured µg/ g | Recovery rate |
|---|---|---|---|---|---|
| Al | 80 mg/g | 92% | Be | 2.99 | 93% |
| Ba | 433 | 90% | Rb | 160 | 107% |
| Ca | 9.3mg/g | 95% | Mo | 1.22 | 76% |
| Fe | 49 mg/g | 102% | Cd | 0.265 | 133% |
| K | 34 mg/g | 114% | Sb | 0.873 | 91% |
| Li | 100 | 126% | Pb | 25.1 | 105% |
| Mg | 19 mg/g | 106% | U | 2.69 | 100% |
| Mn | 784 | 127% | V | 159 | 114% |
| Na | 19 mg/g | 68% | Cr | 103 | 107% |
| P | 826 | 118% | Co | 23 | 116% |
| Sc | 16.7 | 98% | Ni | 80.3 | 152% |
| Sr | 122 | 82% | Cu | 29 | 97% |
| Ti | 3.82 mg/g | 85% | As | 8.31 | 90% |
| Zn | 187 | 144% | La | 44 | 110% |
| Zr | 144 | 111% | Ce | 91 | 109% |
| | | | Pr | 10.7 | 120% |
| | | | Nd | 41 | 115% |
| | | | Sm | 7.7 | 109% |
| | | | Eu | 1.49 | 98% |
| | | | Gd | 6.2 | 104% |
| | | | Tb | 0.87 | 90% |
| | | | Dy | 4.8 | 90% |
| | | | Ho | 0.88 | 82% |
| | | | Er | 2.5 | 103% |
| | | | Tm | 0.34 | 76% |
| | | | Yb | 2.3 | 82% |
| | | | Lu | 0.33 | 76% |

**Table B1.** MAG–1 recovery rates. Elements have a recovery rate between 70% and 130%, except for Zn and Ni. The very low amount of CRM used (<10 mg) could explain the difference observed in recovery rates, because subsampling heterogeneity is possible with such small amounts. Zn and Ni are overestimated, probably due to contamination at trace levels.

## Appendix B: Geostandard recovery rates

Recoveries of geostandards MAG–1 in Table B1 and SCO–1 in Table B2.



| Analysed by ICP-AES | Measured μg/g or mg/g | Recovery rate | Analysed by ICP-MS | Measured μg/g | Recovery rate |
|---|---|---|---|---|---|
| Al | 58.5 mg/g | 81% | Be | 1.87 | 104% |
| Ba | 426 | 75% | Rb | 135 | 123% |
| Ca | 13.1 mg/g | 70% | Mo | 1.46 | 104% |
| Fe | 30.3 mg/g | 84% | Sb | 2.73 | 109% |
| K | 22.1 mg/g | 96% | Pb | 32.6 | 105% |
| Li | 50 | 111% | V | 160 | 123% |
| Mg | 13.1 mg/g | 80% | Cr | 76.3 | 112% |
| Mn | 340 | 83% | Co | 12.7 | 116% |
| Na | 10.1 mg/g | 152% | Ni | 28.6 | 106% |
| P | 827 | 90% | Cu | 31.6 | 109% |
| Sc | 10.06 | 92% | As | 12.2 | 101% |
| Sr | 127 | 75% | La | 32 | 104% |
| Ti | 2.84 | 75% | Ce | 63 | 98% |
| Zn | 117 | 117% | Pr | 7.7 | 110% |
| Zr | 129 | 81% | Nd | 29 | 110% |
| | | | Sm | 5.6 | 102% |
| | | | Eu | 1.20 | 113% |
| | | | Gd | 4.6 | 101% |
| | | | Tb | 0.66 | 87% |
| | | | Dy | 3.8 | 96% |
| | | | Ho | 0.72 | 76% |
| | | | Er | 2.1 | 82% |
| | | | Tm | 0.31 | 72% |
| | | | Yb | 2.0 | 84% |
| | | | Lu | 0.31 | 79% |

**Table B2.** SCO–1 recovery rates. Elements, except Na (RR%= 150%), have a recovery rate between 70% and 130%. The very low amount of CRM used (<10 mg) could explain the difference observed in recovery rates, because subsampling heterogeneity is possible with such small amounts.





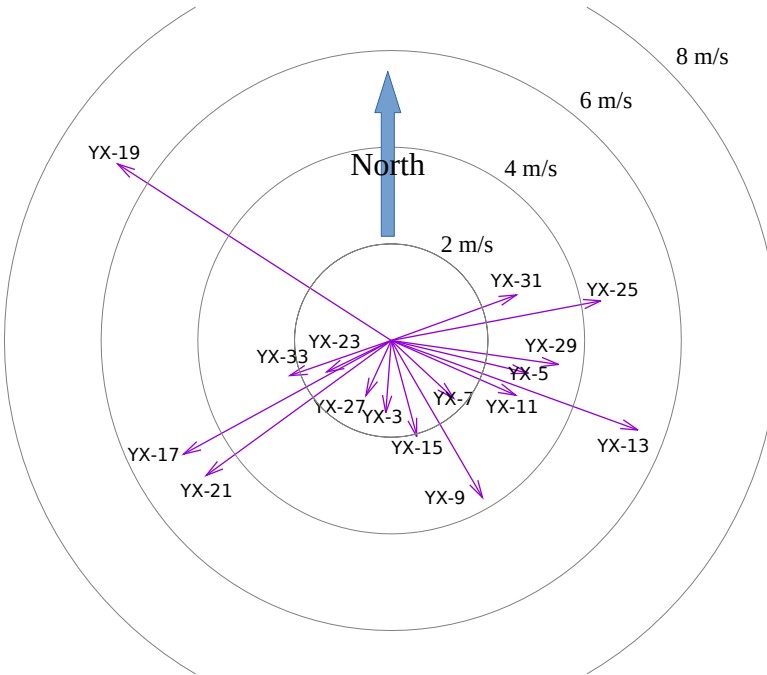

**Figure C1.** Vector representing local wind conditions at the sampling station. The length of the vector represents wind speed average and its angle indicates the average direction during each sampling period.

**Appendix C:  Local meteorological conditions and air trajectories**

Wind speed and direction are measured continuously at the sampling location and backward air trajectories are calculated using the on-line facility at NOAA HYSPLIT model web pages (Stein et al., 2015; Rolph et al., 2017). Trajectories for a 24-hour period are calculated every 6 hours (at midnight, 6 AM, noon and 6 PM).

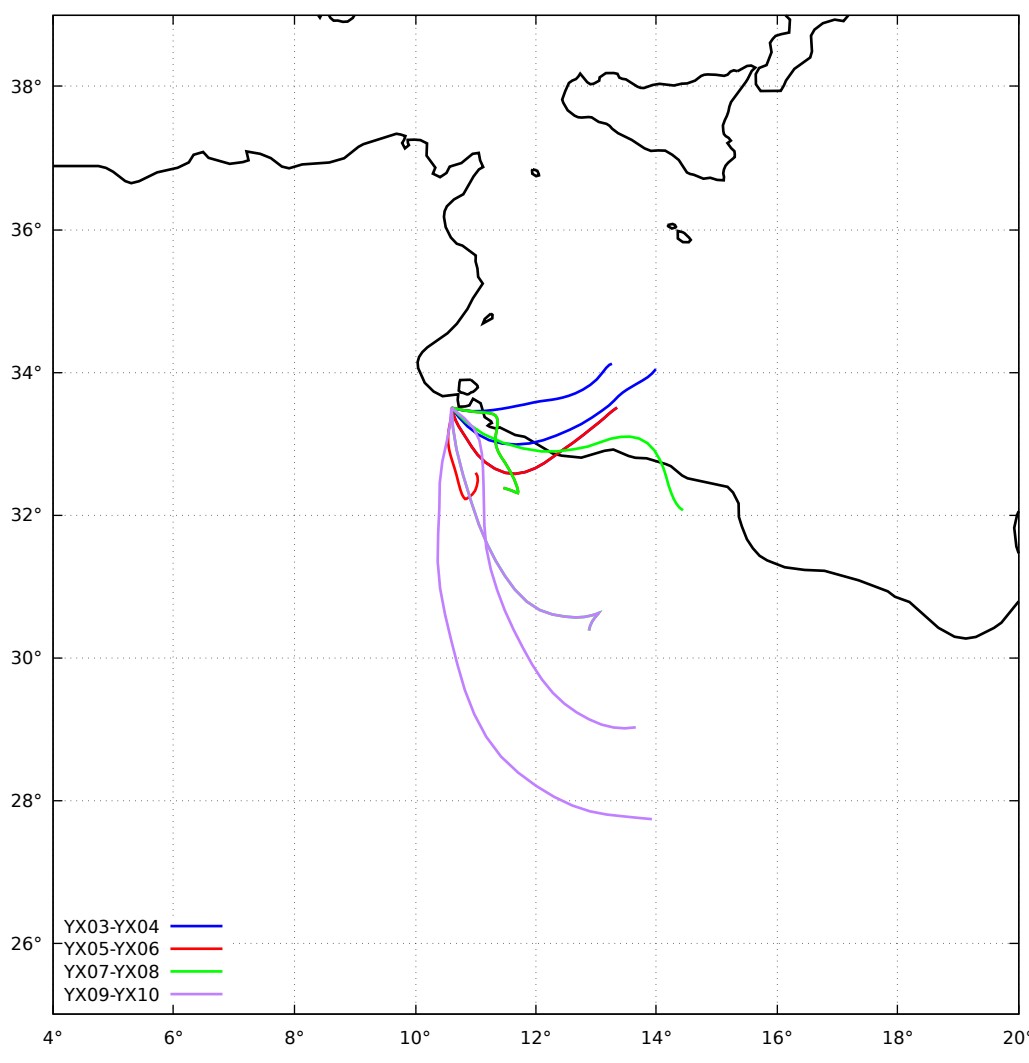

**Figure C2.** Backward-trajectories of sample pairs YX03-YX04, YX05-YX06, YX07-YX08 and YX09-YX10. The x axis is longitude, the y axis is latitude. Two or three trajectories are associated with a given sample pair of ≈12 h duration.

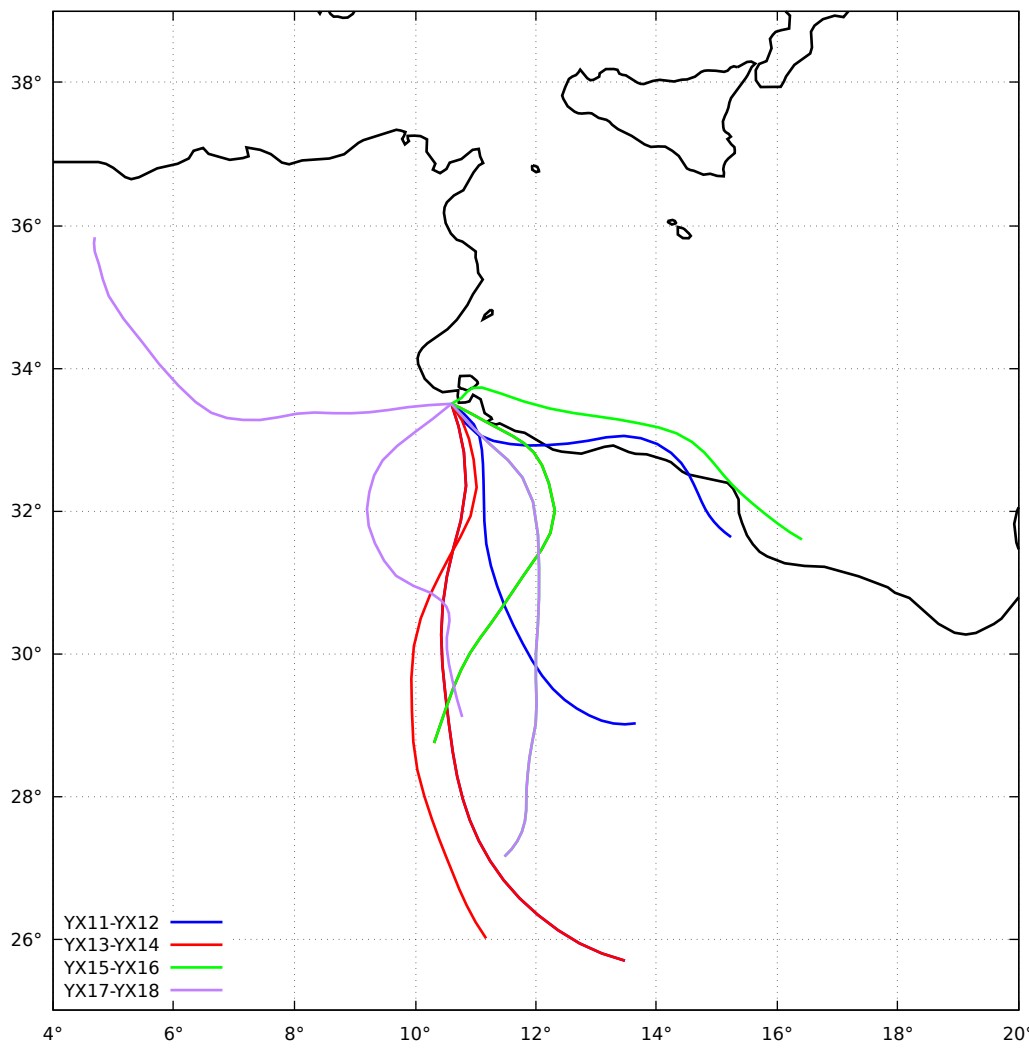

**Figure C3.** Backward-trajectories of sample pairs YX11-YX12, YX13-YX14, YX15-YX16 and YX17-YX18. The x axis is longitude, the y axis is latitude. Two or three trajectories are associated with a given sample pair of ≈12 h duration.

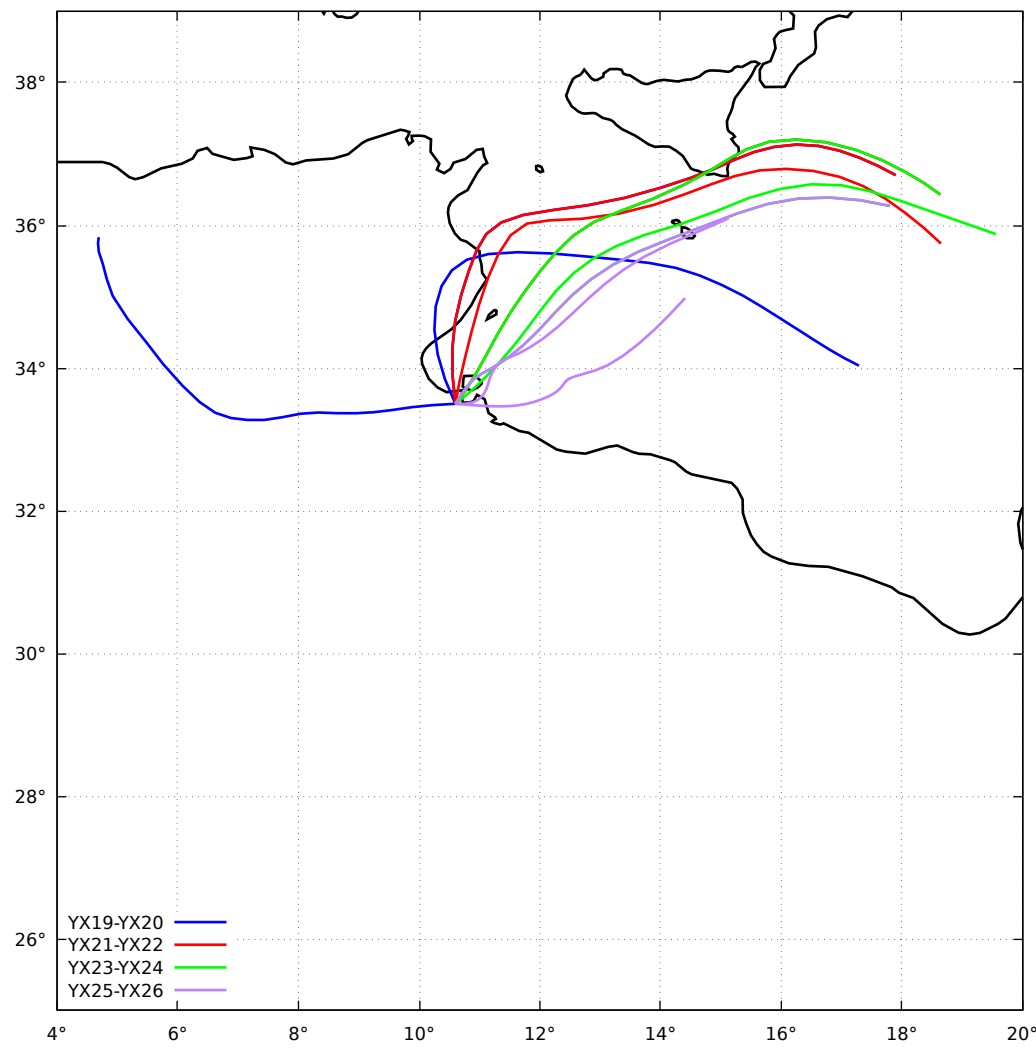

**Figure C4.** Backward-trajectories of sample pairs YX19-YX20, YX21-YX22, YX23-YX24 and YX25-YX26. The x axis is longitude, the y axis is latitude. Two or three trajectories are associated with a given sample pair of ≈12 h duration.



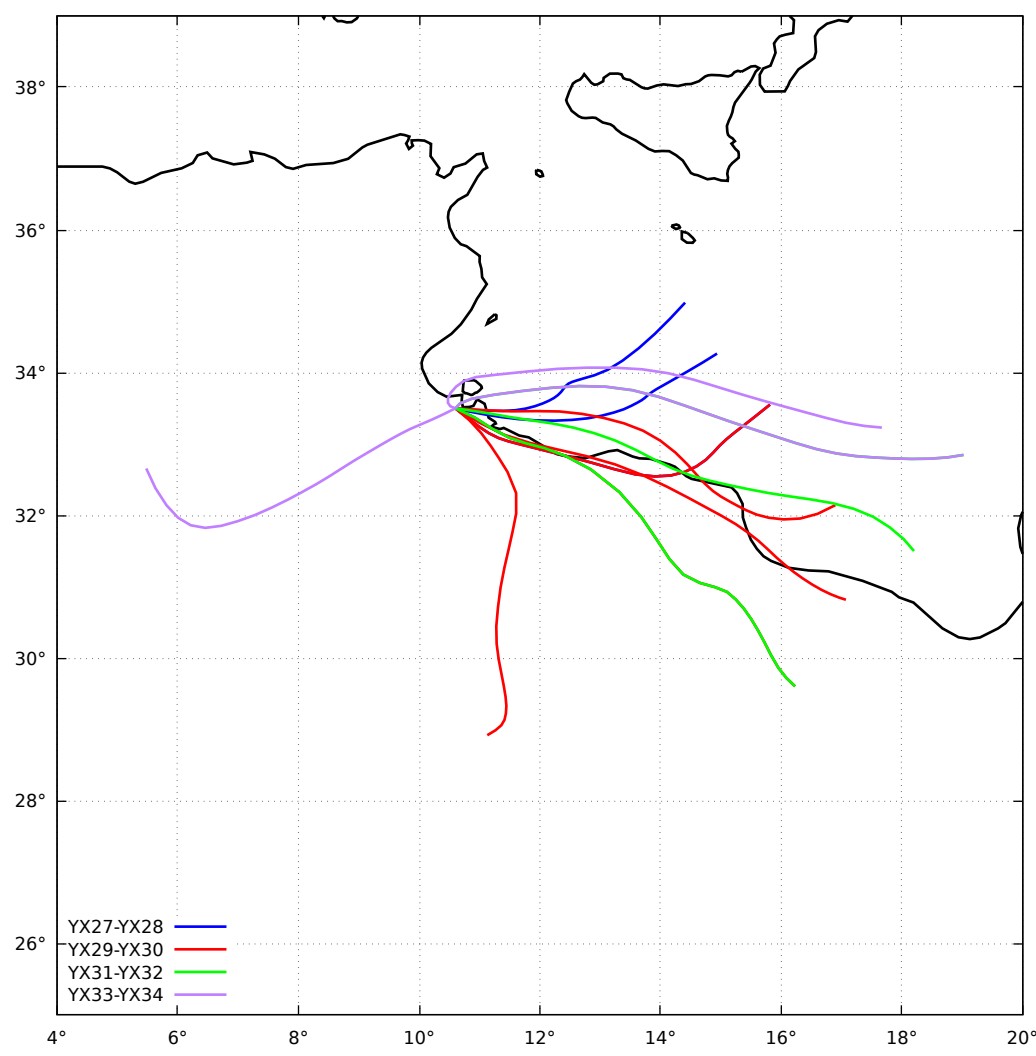

**Figure C5.** Backward-trajectories of sample pairs YX27-YX28, YX29-YX30, YX31-YX32 and YX33-YX34. The x axis is longitude, the y axis is latitude. Two or three trajectories are associated with a given sample pair of ≈12 h duration. except for the pair YX29-YX30 for which four trajectories are necessary because the sampling duration was 24 hours.



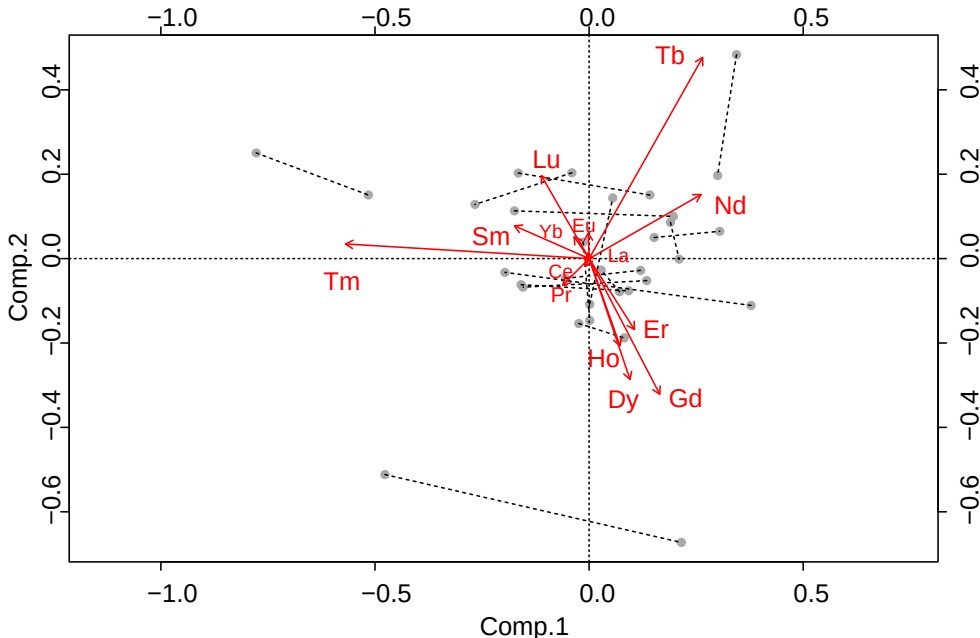

**Figure D1.** REE Biplot for two VTD simulation results. Percentages of variability explained by the first two components are 27% and 19%, a total of 46%.

**Appendix D: REE biplot simulated with the observed analytical uncertainty.**



*Author contributions.* All the authors contributed to this paper to an extent reflected by their rank in the authors list

*Competing interests.* No competing interest are present

*Acknowledgements.* This work was supported by NILDAE (Nutrients Inputs and Losses Due to Aeolian Erosion in the Sahel), a program of Université Sorbonne Paris Cité. We thank the Institut des Régions Arides (IRA) for providing logistics and support during the experiment. Special thanks are expressed to T.H. des Tureaux for his invaluable help during sampling in Tunisia, and to E. Bon Nguyen, M. Tharaud, A.
Feron, J. Chane Teng, and Z. Qu for their advice and help during the analyses. Thanks to Carmela Chateau-Smith who reviewed the English.





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
