# Peer review of "Compositional data analysis (CoDA) as a tool to evaluate a new low-cost settling-based PM10 sampling head in a desert dust source region"

_Atmospheric Measurement Techniques, 2021_

## Referee Comment (RC2)

**Review: Compositional data analysis (CoDA) as a tool to evaluate a new low-cost settling-based PM$_{10}$ sampling head in a desert dust source region**

This paper presents a novel design of PM$_{10}$ sample inlet and a compositional method to compare it to an aluminum alloy commercial PM$_{10}$ impactor inlet. The authors aim to show that vertical tube decanter can replace standard, impaction-based PM$_{10}$ sampling head. While compositional data analysis shows no significant differences between sampling systems, the mass collection efficiency is only partially analyzed, which is the main weakness of the current paper.

1. Size-selective PM sampling inlets play an important role in ambient PM measurements. The main motivation for developing a new PM$_{10}$ sample inlet presented in this paper is the elimination of the contamination of aerosol samples with metal particles (friction between coarse particles and metallic parts of the standard PM10 inlet system).
    a. Was such contamination observed in any other study (references needed)?
    b. The results of this study indicate that there is no such contamination using a standard PM10 sample system. Hence, the authors should justify the study more clearly. One possible reason to use a vertical tube decanter (VTD) as a PM10 inlet is easier maintenance. It is well known that PM10 impactor inlet systems must be cleaned regularly. Deposited particles that do not stick well on the impaction surface can be deagglomerated and re-entrained to the downstream, leading to oversampling. See (Le et al., 2019) and references therein. However, the disadvantages of using a simplified system like VTD must be discussed as well (see point 3.)

2. Before intercomparison of the chemical composition of particles sampled with both inlets, authors should thoroughly compare the total mass of PM10 measured by the three sampling systems. The conclusion such as (p.10, line 180): " *To summarize, the differences observed between aerosol masses measured by the three sampling systems are much lower than the daily variability observed during the field experiment.*" is not adequate. The intercomparison should be done in two steps; firstly, to compare "crustal composition method for determination of aerosol total mass" for filters using standard PM10 inlet to reference gravimetric method (TEOM), and secondly, to compare VDT and standard PM10 inlets both using "crustal composition method". One way to show this "indirect equivalence" is following the tools and methods developed in standard EN16450:2017 (EN 16450:2017, 2017). The reference method for the first step is defined in EN 12341:2014 (EN 12341:2014, 2014). Nevertheless, a proper application of EN16450:2017 requires a minimum of 40 valid data pairs with the further requirement of two candidate applications for each type of testing application. Additionally, the same standard further describes requirements related to the number of locations and the concentration range of data points.
    However, authors should at least perform an orthogonal regression algorithm for both steps and comment slope, intercept, and variances of the intercomparison results. The authors should update Figure 6 accordingly.
    a. The conclusion such as (p. 16, line 220): "*Consequently, both devices can be used for the determination of mass and chemical composition of220aerosols in source regions,*

> *or even simply to determine mass by gravimetry.*" is true only if the equivalence is proven.

  b. Quick orthogonal regression intercomparison of "crustal composition method for determination of aerosol total mass" for filters using standard PM10 inlet to reference gravimetric method (TEOM) in the range up to 115 µg/m$^3$shows slope lower than 0.9 and significant intercept. Authors should comment on the uncertainties of aluminum sea-salt sodium and crust sodium ratios used in the crust model for the total aerosol mass for the specific location.

  c. Is the assumption of neglecting the organic molecules in the model accurate for the lower mass concentration range (possible secondary organic aerosol formation)?

  d. Authors should show mass concentration size distribution (measured using GRIMM OPC) for a low concentration regime (Figure 5) as well; day March 30 2016, for example.

3. It would be interesting to compare the VTD cut-off curve to the standard PM10 inlet cut-off curve. If available, authors should plot both in Figure 4.

  a. From the Figure 4. It can be seen that cut-off diameter for a cylinder system with a diameter of 125 mm is approx. 14 µm at 17 LPM and not 10 µm?

  b. What is the length of the VTD, and does it play any role? Why did you choose the specific VTD length?

  c. Can you comment on the influence of wind speed on VTD sampling efficiency? For example, see (Lee et al., 2013; Faulkner et al., 2014) and references therein.

4. The caption for Figure 3 is not adequate. Authors should describe subpictures (a), (b), and (c) in detail.

5. P. 12, line 197. Do you mean perturbation vector VTD instead of VTP?

**References**

EN 12341:2014: Ambient air - Standard gravimetric measurement method for the determination of the PM10 or PM2.5 mass concentration of suspended particulate matter, European committee for standardization, Brussels, Netherlands, 2014.

EN 16450:2017: Ambient air - Automated measuring systems for the measurement of concentration of particulate matter (PM10; PM2.5), European committee for standardization, Brussels, Netherlands, 2017.

Faulkner, W. B., Smith, R., and Haglund, J.: Large Particle Penetration During PM $_{10}$ Sampling, Aerosol Science and Technology, 48, 676–687, https://doi.org/10.1080/02786826.2014.915005, 2014.

Le, T.-C., Shukla, K. K., Sung, J.-C., Li, Z., Yeh, H., Huang, W., and Tsai, C.-J.: Sampling efficiency of low-volume PM $_{10}$ inlets with different impaction substrates, Aerosol Science and Technology, 53, 295–308, https://doi.org/10.1080/02786826.2018.1559919, 2019.

Lee, S., Yu, M., and Kim, H. H.: Development of aerosol wind tunnel and its application for evaluating the performance of ambient PM10 inlets, Atmospheric Pollution Research, 4, 323–328, https://doi.org/10.5094/APR.2013.036, 2013.

---

## Author Comment (AC2)

**Responses from the authors to Reviewer #2 comments:**

This paper presents a novel design of PM10 sample inlet and a compositional method to compare it to an aluminum alloy commercial PM10 impactor inlet. The authors aim to show that vertical tube decanter can replace standard, impaction-based PM10 sampling head. While compositional data analysis shows no significant differences between sampling systems, the mass collection efficiency is only partially analyzed, which is the main weakness of the current paper.

**1.** Size-selective PM sampling inlets play an important role in ambient PM measurements. The main motivation for developing a new PM10 sample inlet presented in this paper is the elimination of the contamination of aerosol samples with metal particles (friction between coarse particles and metallic parts of the standard PM10 inlet system).

> a. Was such contamination observed in any other study (references needed)?

> **Answer:** we did not find any published papers directly addressing such issues. However, because friction of dust particles on impaction plates of "standard" PM10 inlet systems is a potential source of contamination (essentially Al and Mg, plus traces like Zn and Cu in case of aluminium alloy), a study on potential contamination is interesting. As mentioned in b) below, another contamination issue is the accumulation of previous sampling on the impaction plate and bouncing. One conclusion of our study is that we have not detected such contamination for the brand new commercial PM10 used.

> b. The results of this study indicate that there is no such contamination using a standard PM10 sample system. Hence, the authors should justify the study more clearly. One possible reason to use a vertical tube decanter (VTD) as a PM10 inlet is easier maintenance. It is well known that PM10 impactor inlet systems must be cleaned regularly. Deposited particles that do not stick well on the impaction surface can be deagglomerated and re-entrained to the downstream, leading to oversampling. See (Le et al., 2019) and references therein. However, the disadvantages of using a simplified system like VTD must be discussed as well (see point 3.)

> **Answer:** we thank Reviewer #2 for this suggestion and agree, we will add the potential advantages of VTD in the text body.

**2.** Before intercomparison of the chemical composition of particles sampled with both inlets, authors should thoroughly compare the total mass of PM10 measured by the three sampling systems. The conclusion such as (p.10, line 180): " To summarize, the differences observed between aerosol masses measured by the three sampling systems are much lower than the daily variability observed during the field experiment." is not adequate. The intercomparison should be done in two steps; firstly, to compare "crustal composition method for determination of aerosol total mass" for filters using standard PM10 inlet to reference gravimetric method (TEOM), and secondly, to compare VDT and standard PM10 inlets both using "crustal composition method". One way to show this "indirect equivalence" is following the tools and methods developed in standard EN16450:2017 (EN 16450:2017, 2017). The reference method for the first step is defined in EN 12341:2014 (EN 12341:2014, 2014). Nevertheless, a proper application of EN16450:2017 requires a minimum of 40 valid data pairs with the further requirement of two candidate applications for each type of testing application. Additionally, the same standard further describes requirements related to the number of locations and the concentration range of data points. However, authors should at least perform an orthogonal regression algorithm for both steps and comment slope, intercept, and variances of the intercomparison results. The authors should update Figure 6 accordingly.

**Answer:** Figure 6a was not readable enough, and will be modified so that both plots (VTD and PM10) are more clearly seen against TEOM (see Reply to Reviewer #1).

We would be very happy to cite standard European methods. However, readers of scientific journals have access to the published literature through open access and institutional subscriptions. Standard European methods are not published in scientific journals nor are they readily available without high fees. Therefore the scientific community has poor access to these publications and citing them remains problematic.

Nevertheless, we agree with Reviewer #2 about adding statistic quantification to figures 6a and 6b. An orthogonal regression, also known as total least square, was performed by treating the variances of x and y symmetrically, with the help of the function "prcomp" implemented in R. Orthogonal regressions were performed twice, with and without the highest point, which could potentially considered as an outlier. The following tables summarize the results and will be added as supplementary material:

Including the heavy loaded sample

|  | Slope (95%) | Intercept (95%), $\mu g.m^{-3}$ |
|---|---|---|
| VTD = f(TEOM) | [0.78, 1.16] | [-18, +15] |
| PM10 = f(TEOM) | [0.79, 1.07] | [-17, +8] |
| VTD = f(PM10) | [0.96, 1.10] | [-1, +9] |

Without the heavy loaded sample

|  | Slope (95%) | Intercept (95%), $\mu g.m^{-3}$ |
|---|---|---|
| VTD = f(TEOM) | [0.76, 1.20] | [-8, +19] |
| PM10 = f(TEOM) | [0.77, 1.11] | [-9, +10] |
| VTD = f(PM10) | [0.94, 1.12] | [-0.2, +11] |

In each case, slope and intercept are not significantly different from 1 and zero, respectively, suggesting that if any bias really occurs, it is too small to be identified with our data. We suggest including in the text the statistical results obtained when including the heavy loaded sample.

a. The conclusion such as (p. 16, line 220): "Consequently, both devices can be used for the determination of mass and chemical composition of aerosols in source regions, or even simply to determine mass by gravimetry." is true only if the equivalence is proven.

**Answer:** The equivalence is suggested using orthogonal regression, as proposed by Reviewer #2. We will modify the sentence: "Consequently our data suggest that both devices can be used for the determination of mass and chemical composition of aerosols in source regions, or even simply to determine mass by gravimetry."

b. Quick orthogonal regression intercomparison of "crustal composition method for determination of aerosol total mass" for filters using standard PM10 inlet to reference gravimetric method (TEOM) in the range up to 115 µg/m3shows slope lower than 0.9 and significant intercept. Authors should comment on the uncertainties of aluminum sea-salt sodium and crust sodium ratios used in the crust model for the total aerosol mass for the specific location.

**Answer:** see above for the orthogonal regression. The reference used for aluminum (Bowen, 1966) could appear poor because large uncertainties are found for aluminum in soils. A more recent reference stating an aluminum proportion of 7.09% in Saharan dust will be added:

Guieu, C., Loÿe-Pilot, M.-D., Ridame, C., and Thomas, C., Chemical characterization of the Saharan dust end-member: Some biogeochemical implications for the western Mediterranean Sea, *J. Geophys. Res.,* 107( D15), doi:10.1029/2001JD000582, 2002.

c. Is the assumption of neglecting the organic molecules in the model accurate for the lower mass concentration range (possible secondary organic aerosol formation)?

**Answer:** This assumption becomes less accurate if other sources than measured inorganics contribute to the mass. It is the reason why we performed our experiment in an arid region with low organic sources (very few plants, little anthropogenic activity). Ammonium and other molecules containing nitrogen and salts are supposed to be low enough for a proper total mass calculation with our measurements. This explanation will be added in the future version of the text.

d. Authors should show mass concentration size distribution (measured using GRIMM OPC) for a low concentration regime (Figure 5) as well; day March 30 2016, for example.

**Answer:** On March 30, OPC data are measured only during a short period of 2 min at 10:23 AM and the number of large particles counted is not large enough. We propose to add two graphs to the existing figure 5, one at a lower concentration regime, where OPC measurements were recorded for three hours in the evening on March 31, and a second one, at the highest concentration regime, which was measured on April 2 during the largest dust burst. All these graphs will be displayed in mass distribution frequency instead of mass concentration in air.

Future Figure 5a: mass distribution frequency on March 31 evening, aerosol concentration ca. 40 µg.m$^{-3}$.

[Figure]

Future Figure 5b, already present as Figure 5: mass distribution frequency on April 6 day, aerosol concentration ca. 100 µg.m$^{-3}$.

[Figure]

Future Figure 5c: mass distribution frequency on April 2 between 11 AM and 1 PM during the highest dust episode, aerosol concentration ca. 1000 µg.m⁻³.

[Figure]

**3.** It would be interesting to compare the VTD cut-off curve to the standard PM10 inlet cut-off curve. If available, authors should plot both in Figure 4.

**Answer:** This information is not available for the commercial PM10 used, maybe because changes in the aerodynamic conditions around the impaction nose of the sampling head would also greatly disturb performance efficiency .

a. From the Figure 4. It can be seen that cut-off diameter for a cylinder system with a diameter of 125 mm is approx. 14 µm at 17 LPM and not 10 µm?

**Answer:** This was a confusing mistake. A complete answer is provided in responses to Reviewer #1. Briefly, we wanted to indicate that sampling rates are in the range of one cubic meter per hour. Actually, VTD operated at ca. 11 L.min⁻¹. We will modify Figure 4 adding an indication on the flow-rate obtained during our experiments (see reply to Reviewer #1).

b. What is the length of the VTD, and does it play any role? Why did you choose the specific VTD length?

**Answer:** It is commonly admitted that air flow in an open tube is no longer disturbed by the entrance conditions after a distance larger than three times its diameter. For a 125 mm diameter, the minimum length is 375 mm. We have chosen a 500 mm length tube, because it is the closest commercially available length we have found.

c. Can you comment on the influence of wind speed on VTD sampling efficiency? For example, see (Lee et al., 2013; Faulkner et al., 2014) and references therein.

**Answer:** During the two-week experiment, various wind conditions were experienced, as shown in Figure C1. Aitchison distance in the compositional dataset was used as a proxy for compositional differences between the two sampling heads. As can be seen in the figure below, which represents this distance as a function of wind speed, no significant linear or monotone dependence was found using Pearson (p-value = 0.47) and Spearman (p-value = 0.35) correlation coefficients, respectively. A sentence will be added, stating that the slight differences observed with the two sampling heads are independent of wind speed.

[Figure]

If the ratios between calculated VTD mass and measured TEOM mass are plotted as a function of wind speed, no correlation is observed (figure below), with Pearson and Spearman p-values equal to 0.98 and 0.76, respectively.

[Figure]

A sentence will be added in the text body to summarize these conclusions without adding these figures.

**4.** The caption for Figure 3 is not adequate. Authors should describe subpictures (a), (b), and (c) in detail.

**Answer:** We agree with Reviewer #2, we will provide more details in the figure caption.

**5.** P. 12, line 197. Do you mean perturbation vector VTD instead of VTP?

**Answer:** Yes, there is a typo in the formula. VTD should be read instead of VTP, a mistake is also present a few lines above where VDT is written instead of VTD.

==========

**References**

EN 12341:2014: Ambient air -Standard gravimetric measurement method for the determination of the PM10 or PM2.5 mass concentration of suspended particulate matter, European committee for standardization, Brussels, Netherlands, 2014.

EN 16450:2017: Ambient air -Automated measuring systems for the measurement of concentration of particulate matter (PM10; PM2.5), European committee for standardization, Brussels, Netherlands, 2017.

Faulkner, W. B., Smith, R., and Haglund, J.: Large Particle Penetration During PM 10Sampling, Aerosol Science and Technology, 48, 676–687, https://doi.org/10.1080/02786826.2014.915005, 2014.

Le, T.-C., Shukla, K. K., Sung, J.-C., Li, Z., Yeh, H., Huang, W., and Tsai, C.-J.: Sampling efficiency of low-volume PM 10inlets with different impaction substrates, Aerosol Science and Technology, 53, 295–308, https://doi.org/10.1080/02786826.2018.1559919, 2019.

Lee, S., Yu, M., and Kim, H. H.: Development of aerosol wind tunnel and its application for evaluating the performance of ambient PM10 inlets, Atmospheric Pollution Research, 4, 323–328, https://doi.org/10.5094/APR.2013.036, 2013.

---

## Author Response (AR1)

**Responses to Reviewers**

We thank the Reviewers for their comments that have greatly contributed to improve this paper. We have carefully answered to all the addressed question and also we have made some other minor changes to present this new version. A summary of the changes can be found in the "differential document".

**Responses from the authors to Reviewer #1 comments:**

The use of "low cost" $PM_{10}$ heads can be of interest in desert dust source areas to chemically characterize mineral dust and for further evaluation of PM10 levels. The manuscript presents a new PM10 sampling head and discuss the use of Compositional data analysis (CoDA) for the performance evaluation of the new inlet.

The main objective of this paper is to show that a low-cost decanter tube can replace an impaction-based $PM_{10}$ sampling head for proper aerosol sampling. However, this objective is not exactly reflected in the tittle that focus on the use of the CoDA as a tool to evaluate a new low-cost settling-based $PM_{10}$. The objectives of this paper should be clearly stated at the end of the introductory section.

**Answer:** We agree with this comment, we have moved the sentence "The objective of this paper is to show that a low-cost decanter tube can replace an impaction-based PM10 sampling head for proper aerosol sampling." (former lines 51-52), to the end of the paragraph, and add a few words on an application of CoDA, the innovative new tool for data analysis in order to emphasise both of the objectives of this study (line 61):

1 Major objective: evaluation of a new low-cost decanter tube

2 Minor objective: use of CoDA (based on log ratios) to compare results using a robust method of data analysis.

The interest of a new head for PM10 sampling at source areas should be justified. The manuscript concluded that both the new inlet and the commercial one can be indistinctly used. However, the advantages of the new inlet are not clearly justified.

**Answer:** The main advantage of this new inlet is its simple design associated with its low cost and the broad availability of the components. This new inlet can be build by everyone with local materials. We have added this sentence to the paper and also have added another advantage on maintenance suggested by Reviewer#2 (line 237).

As stated by the authors, differences on the chemical compositions of samples collected simultaneously by both VDT and commercial PM10 heads may differ due to "contamination, size segregation of particles, and mineralogical fractionation during sampling". Thus, one of the reasons of the using the new VDT sampler is the potential contamination of the sample when using the PM10 commercial due to wear of the metal impact plate. The results show similarity between the two samplers and appear to indicate that there is no contamination. Have you observed a contamination of the PM10 sample by Al due to the friction of the particles with the aluminum plate? Regarding the chemical composition of the filters collected with both samplers and presented in the tables, it seems that there is no enrichment in Al in those filters collected with the commercial entry of PM10. Therefore, the contamination cannot be confirmed.

**Answer:** Yes, potential contamination issues due to aluminium impaction plates were among the main reasons why we test sampling heads in the field. No compositional differences were observed between the two sampling heads (although they are made of different alloys). This factor strongly suggests that neither device would contaminate natural samples. This have been stated in more detail in this version of the manuscript (line 219, see also answer to Reviewer#2).

The methods used in the article for the evaluation of PM10 inlet are adequate. However, it would have been interesting to make the comparison of PM10 concentrations directly from gravimetric determinations, in addition to the comparison of the chemical composition.

**Answer:** Aerosols for further chemical analyses were sampled using cellulose ester filters, which are not suitable for weighing because of it moisture sensitivity. This potential issue was anticipated. That is the reason why a TEOM was installed, as it directly provides aerosol aerosol mass concentration in air. We have added a sentence in the text stating that we have anticipated this issue (line 130).

The concept "Compositional data analysis (CoDA)" only appears in the Title and conclusions sections. It should be also mentioned in the introduction section, at least, as one of the objectives of the paper and in the methodology section. The acronym "CoDA" is only used in the Title.

**Answer**: We thank the reviewer for identifying this problem. The acronym 'CoDA' is properly defined in this version and recalled in the text at the places it should be.

It is concluded in the paper that there are not differences were evidenced for samples collected near a source region. I understand you refer to a source region of mineral dust. Please, specify this in the text.

**Answer**: Reviewer #1 is right. The Conclusions based on our experiments were for a source region of mineral dust. We have added this notice in the manuscript (lines 242-246).

Have you evidenced differences between the two samplers for low and high concentrations of PM?

**Answer**: The differences in aerosol mass concentration observed between the two sampling heads were always small and, importantly, independent of the level of PM10 concentration in the air, although that level varied greatly, from 25 to 700 µg/m$^3$ (Figure 6). Euclidean distance in the compositional biplot (PC1-PC2 projection, Figure 7a) was used as a proxy for compositional differences between the two sampling heads. As can be seen in the figure below, which represents this distance as a function of PM10 concentrations, the two parameters are independent. Note that these values of Euclidian distances below 0.7 are small in comparison with the range within PC1 (~7) and PC2 (~4). A sentence have been added, stating that the slight differences observed with both sampling heads are independent on air aerosol concentrations (line 218, see also answer to Reviewer#2).

[Figure]

for reviewers, not to be published

Does the shape of particles (somehow related to the mineralogical, and therefore chemical, composition) affect the behaviour of the samplers?

**Answer:** As no mineralogical study was performed, we cannot fully address this question. However, the location of the sampling site (not far from the ocean and located in a desert region that is sometimes impacted by local anthropogenic activities, such as biomass burning) allowed us to have samples with very different mineralogical and chemical compositions. If there is an impact of mineralogy, it has no noticeable deleterious influence on chemical composition, as no differences were observed between the two sampling heads (see answers above).

Can you confirm that the PM10 head shown in figure 1 is Tecora PM10, as mentioned in the text?

**Answer:** Yes, both PM10 and TEOM heads are from the same brand: Tecora PM10. We have added this information in the legend of Figure 1.

The acronym PLAS is not frequently used; I would prefer to use OPC (optical particle counter) or OPS (optical particles spectrometer)

**Answer:** Yes, Reviewer #1 is right, we have changed the text to replace PLAS by OPC.

line 133: "[particles]air"; do you mean crustal particles??

**Answer:** Yes, it is a mistake, we mean "crustal particles", this has been corrected in this version.

Line 165: the concentrations reported here refer to PM10 or PM25? Please, specify that are concentrations of sea salt (and crustal) in PM10.

**Answer:** We refer to "chemically weighed" PM10 particles

Table 1. Please, add the same information (concentration of crustal and sea salt fraction and Ca species) for samples collected with the commercial PM10 head. You can add the info in Table 1 or in the supplementary. Please, unify the criterion for the number of decimal places - (the same for Tables A1 to A7).

**Answer:** A larger table in the Appendix E (Tables E1 and E2), including 3 columns plus the sum for each type of sampling head is now provided, following this remark by Reviewer #1. For a better visibility, we have presented Sea salt and Crustal fraction percentages in table 1.

Figure 4: According to figure 4 the flow for PM10 cut is close to 10 L min-1; However, the flow used was 17 l min-1.

**Answer:** Figure 4 refers to VTD and not PM10. The legend was not probably clear enough and has been changed, to indicate VTD. We also see that line 4 in the abstract and line 58 in the manuscript are not well written and could be confusing. We wanted to indicate that sampling rates are in the range of one cubic meter per hour samplers. The sentence have been changed to directly indicate "in the range of one cubic meter per hour". We have also modified the end of former line 62, writing "operating within the same flow rate range" (line 63) instead of "operating at the same flow rate" and in other parts of the manuscript we have replaced 17 L.min$^{-1}$ by 1 m$^3$.hour$^{-1}$.

VTD operated at ca. 11 L.min$^{-1}$ and a grey dot is added in figure 4 to indicate the operating conditions, as shown below.

[Figure]

modified figure 4

Figure 6a: please, change colours and/or shapes of dots. Difficult to discriminate.

**Answer:** The new figure is drawn black and white using suitable symbols (see below).

[Figure]

modified figure 6a

**Responses from the authors to Reviewer #2 comments:**

This paper presents a novel design of PM10 sample inlet and a compositional method to compare it to an aluminum alloy commercial PM10 impactor inlet. The authors aim to show that vertical tube decanter can replace standard, impaction-based PM10 sampling head. While compositional data analysis shows no significant differences between sampling systems, the mass collection efficiency is only partially analyzed, which is the main weakness of the current paper.

**1.** Size-selective PM sampling inlets play an important role in ambient PM measurements. The main motivation for developing a new PM10 sample inlet presented in this paper is the elimination of the contamination of aerosol samples with metal particles (friction between coarse particles and metallic parts of the standard PM10 inlet system).

> a. Was such contamination observed in any other study (references needed)?
>
> **Answer:** we did not find any published papers directly addressing such issues. However, because friction of dust particles on impaction plates of "standard" PM10 inlet systems is a potential source of contamination (essentially Al and Mg, plus traces like Zn and Cu in case of aluminium alloy), a study on potential contamination is interesting. As mentioned in b) below, another contamination issue is the accumulation of previous sampling on the impaction plate and bouncing. One conclusion of our study is that we have not detected such contamination for the brand new commercial PM10 used. We have added this notice in the text (line 219). However, as an extension of this question, we have found publications about sampling head efficiency comparison and added two references (line 192).
>
> b. The results of this study indicate that there is no such contamination using a standard PM10 sample system. Hence, the authors should justify the study more clearly. One possible reason to use a vertical tube decanter (VTD) as a PM10 inlet is easier maintenance. It is well known that PM10 impactor inlet systems must be cleaned regularly. Deposited particles that do not stick well on the impaction surface can be deagglomerated and re-entrained to the downstream, leading to oversampling. See (Le et al., 2019) and references therein. However, the disadvantages of using a simplified system like VTD must be discussed as well (see point 3.)
>
> **Answer:** we thank Reviewer #2 for this suggestion and agree, we have added the potential advantages of VTD in the text body (line 42).

**2.** Before intercomparison of the chemical composition of particles sampled with both inlets, authors should thoroughly compare the total mass of PM10 measured by the three sampling systems. The conclusion such as (p.10, line 180): " To summarize, the differences observed between aerosol masses measured by the three sampling systems are much lower than the daily variability observed during the field experiment." is not adequate. The intercomparison should be done in two steps; firstly, to compare "crustal composition method for determination of aerosol total mass" for filters using standard PM10 inlet to reference gravimetric method (TEOM), and secondly, to compare VDT and standard PM10 inlets both using "crustal composition method". One way to show this "indirect equivalence" is following the tools and methods developed in standard EN16450:2017 (EN 16450:2017, 2017). The reference method for the first step is defined in EN 12341:2014 (EN 12341:2014, 2014). Nevertheless, a proper application of EN16450:2017 requires a minimum of 40 valid data pairs with the further requirement of two candidate applications for each type of testing application. Additionally, the same standard further describes requirements

related to the number of locations and the concentration range of data points. However, authors should at least perform an orthogonal regression algorithm for both steps and comment slope, intercept, and variances of the intercomparison results. The authors should update Figure 6 accordingly.

**Answer:** Figure 6a was not readable enough, and was modified so that both plots (VTD and PM10) are more clearly seen against TEOM (see Reply to Reviewer #1).

We would be very happy to cite standard European methods. However, standard European methods are not published in scientific journals and are available only with high fees. Therefore the scientific community has poor access to these publications and citing them remains problematic. This is a real issue for the efforts made by the European community to create standard methods while US standards are freely available.

Nevertheless, we agree with Reviewer #2 about adding statistic quantification to figures 6a and 6b. An orthogonal regression, also known as total least square, was performed by treating the variances of x and y symmetrically, with the help of the function "prcomp" implemented in R. Orthogonal regressions were performed twice, with and without the highest point, which could potentially considered as an outlier. The following tables summarize the results and were added as supplementary material:

Including the heavy loaded sample

|  | Slope (95%) | Intercept (95%), $\mu g.m^{-3}$ |
|---|---|---|
| **VTD = f(TEOM)** | [0.78, 1.16] | [-18, +15] |
| **PM10 = f(TEOM)** | [0.79, 1.07] | [-17, +8] |
| **VTD = f(PM10)** | [0.96, 1.10] | [-1, +9] |

Without the heavy loaded sample

|  | Slope (95%) | Intercept (95%), $\mu g.m^{-3}$ |
|---|---|---|
| **VTD = f(TEOM)** | [0.76, 1.20] | [-8, +19] |
| **PM10 = f(TEOM)** | [0.77, 1.11] | [-9, +10] |
| **VTD = f(PM10)** | [0.94, 1.12] | [-0.2, +11] |

In each case, slope and intercept are not significantly different from 1 and zero, respectively, suggesting that if any bias really occurs, it is too small to be identified with our data (line 194).

a. The conclusion such as (p. 16, line 220): "Consequently, both devices can be used for the determination of mass and chemical composition of aerosols in source regions, or even simply to determine mass by gravimetry." is true only if the equivalence is proven.

**Answer:** The equivalence is suggested using orthogonal regression, as proposed by Reviewer #2. We have modified the sentence: "Consequently our data suggest that both devices can be used for the determination of mass and chemical composition of aerosols in source regions, or even simply to determine mass by gravimetry." (line 245)

b. Quick orthogonal regression intercomparison of "crustal composition method for determination of aerosol total mass" for filters using standard PM10 inlet to reference gravimetric method (TEOM) in the range up to 115 µg/m3shows slope lower than 0.9 and significant intercept. Authors should comment on the uncertainties of aluminum sea-salt

sodium and crust sodium ratios used in the crust model for the total aerosol mass for the specific location.

**Answer:** see above for the orthogonal regression. The reference used for aluminium (Bowen, 1966) could appear poor because large uncertainties are found for aluminium in soils. A more recent reference stating an aluminium proportion of 7.09% in Saharan dust has been added (line 139):

Guieu, C., Loÿe-Pilot, M.-D., Ridame, C., and Thomas, C., Chemical characterization of the Saharan dust end-member: Some biogeochemical implications for the western Mediterranean Sea, *J. Geophys. Res.*, 107( D15), doi:10.1029/2001JD000582, 2002.

c. Is the assumption of neglecting the organic molecules in the model accurate for the lower mass concentration range (possible secondary organic aerosol formation)?

**Answer:** This assumption becomes less accurate if other sources than measured inorganics contribute to the mass. It is the reason why we performed our experiment in an arid region with low organic sources (very few plants, little anthropogenic activity). Ammonium and other molecules containing nitrogen and salts are supposed to be low enough for a proper total mass calculation with our measurements. This explanation have been added to the text (line 200).

d. Authors should show mass concentration size distribution (measured using GRIMM OPC) for a low concentration regime (Figure 5) as well; day March 30 2016, for example.

**Answer:** On March 30, OPC data are measured only during a short period of 2 min at 10:23 AM and the number of large particles counted is not large enough. We have added two graphs to the existing figure 5, one at a lower concentration regime, where OPC measurements were recorded for three hours in the evening on March 31, and a second one, at the highest concentration regime, which was measured on April 2 during the largest dust burst. All these graphs are now displayed in mass distribution frequency instead of mass concentration in air (Figures 5a, 5b, 5c).

**3.** It would be interesting to compare the VTD cut-off curve to the standard PM10 inlet cut-off curve. If available, authors should plot both in Figure 4.

**Answer:** This information is not available for the commercial PM10 used, maybe because changes in the aerodynamic conditions around the impaction nose of the sampling head would also greatly disturb performance efficiency and the commercial PM10 head used must be run at its nominal flow rate only.

a. From the Figure 4. It can be seen that cut-off diameter for a cylinder system with a diameter of 125 mm is approx. 14 μm at 17 LPM and not 10 μm?

**Answer:** This was a confusing mistake. A complete answer is provided in responses to Reviewer #1. Briefly, we wanted to indicate that sampling rates are in the range of one cubic meter per hour. Actually, VTD operated at ca. 11 L.min$^{-1}$. We have modified Figure 4 adding an indication on the flow-rate obtained during our experiments (see reply to Reviewer #1).

b. What is the length of the VTD, and does it play any role? Why did you choose the specific VTD length?

**Answer:** It is commonly admitted that air flow in an open tube is no longer disturbed by the entrance conditions after a distance larger than three times its diameter. For a 125 mm diameter, the minimum length is 375 mm. We have chosen a 500 mm length tube, because it is the closest commercially available length we have found.

c. Can you comment on the influence of wind speed on VTD sampling efficiency? For example, see (Lee et al., 2013; Faulkner et al., 2014) and references therein.

**Answer:** During the two-week experiment, various wind conditions were experienced, as shown in Figure C1. Aitchison distance in the compositional dataset was used as a proxy for compositional differences between the two sampling heads. As can be seen in the figure below, which represents this distance as a function of wind speed, no significant linear or monotone dependence was found using Pearson (p-value = 0.47) and Spearman (p-value = 0.35) correlation coefficients, respectively.

[Figure]

If the ratios between calculated VTD mass and measured TEOM mass are plotted as a function of wind speed, no correlation is observed (figure below), with Pearson and Spearman p-values equal to 0.98 and 0.76, respectively.

[Figure]

A sentence have been added, stating that the slight differences observed with the two sampling heads are independent of wind speed (line 218).

**4.** The caption for Figure 3 is not adequate. Authors should describe subpictures (a), (b), and (c) in detail.

**Answer:** We agree with Reviewer #2, we have provided more details in the figure caption.

**5.** P. 12, line 197. Do you mean perturbation vector VTD instead of VTP?

**Answer:** Yes, there is a typo in the formula. VTD should be read instead of VTP, a mistake is also present a few lines above where VDT is written instead of VTD.

==========

**References added**

Faulkner, W. B., Smith, R., and Haglund, J.: Large Particle Penetration During PM 10Sampling, Aerosol Science and Technology, 48, 676–687, https://doi.org/10.1080/02786826.2014.915005, 2014.

Guieu, C., Loÿe-Pilot, M.-D., Ridame, C., and Thomas, C., Chemical characterization of the Saharan dust end-member: Some biogeochemical implications for the western Mediterranean Sea, *J. Geophys. Res.,* 107( D15), doi:10.1029/2001JD000582, 2002.

Hitzenberger, R., Berner, A., Galambos, Z., Maenhaut, W., Cafmeyer, J., Schwarz, J., Müller, K., Spindler, G., Wieprecht, W., Acker, K., Hillamo, R., and Mäkelä, T.: Intercomparison of methods to measure the mass concentration of the atmospheric aerosol during INTERCOMP2000—influence of instrumentation and size cuts, Atmospheric Environment, 38, 6467–6476, 2004, https://doi.org/https://doi.org/10.1016/j.atmosenv.2004.08.025.

Le, T.-C., Shukla, K. K., Sung, J.-C., Li, Z., Yeh, H., Huang, W., and Tsai, C.-J.: Sampling efficiency of low-volume PM 10inlets with different impaction substrates, Aerosol Science and Technology, 53, 295–308, https://doi.org/10.1080/02786826.2018.1559919, 2019.

Motallebi, N., Taylor, Jr., C. A., Turkiewicz, K., and Croes, B. E.: Particulate Matter in California: Part 1—Intercomparison of Sev-eral PM2.5, PM10–2.5, and PM10 Monitoring Networks, Journal of the Air & Waste Management Association, 53, 1509–1516, https://doi.org/10.1080/10473289.2003.10466322, 2003.

---

## Author Response (AR2)

**Responses from the authors to Reviewer #1 comments:**

p.33, Appendix D: The slope and intercept values should be shown as well (not only confidence interval): slope [95% confidence interval]; intercept [95% confidence interval]

Tables D1 and D2 are modified as requested.

p.12, line 200-209 (Section 3.3). Authors should use the slope and intercept values from Appendix D to justify the sentence "...the slope and intercept are not significantly different from 1 and 0" or use them to comment on possible discrepancies.

Two sentences are modified lines 194 to 198.

"Regression slopes for the three possible combinations (PM10 vs. TEOM, VTD vs. TEOM, and VTD vs. PM10), with and without the highest point, are between 0.94 and 1.03. The value of one is always included in the 95% confidence level interval associated with each slope, and intercepts are not significantly different from zero (see Tables D1 and D2 in Appendix D), suggesting that any potential bias is too small to be identified with our data. "

Because we have concluded that discrepancies are not visible here, we have not added more comments than what was already written on line 200 and 201 on possible biases:

"The coherence between direct measurement of masses (TEOM) and "chemical" weighing shows that substances not taken into account in our chemical budget (ammonium and organic molecules) do not significantly contribute to the total aerosol mass here."